# Adversarial Vulnerability from Interference Between Features in Superposition

**Edward Stevinson** [1]   **Lucas Prieto** [1]   **Melih Barsbey** [1]   **Tolga Birdal** [1]

## Abstract

Why do adversarial examples exist, and why do they transfer between models? Existing explanations appeal to high-dimensional geometry, non-robust patterns in the input, and decision boundary structure, but none provides a representation-level mechanism that explains why specific perturbations succeed and why attacks transfer between models. In this paper, we show that adversarial vulnerability can stem from *efficient* information encoding in neural networks. Specifically, vulnerability can arise from *superposition* – the phenomenon where networks represent more concepts than they have dimensions, forcing non-orthogonal representation and thus interference. This interference causes perturbations targeting one representation to affect others, creating vulnerabilities determined by interference patterns. In synthetic settings with precisely controlled superposition, we establish that superposition *suffices* to create adversarial vulnerability. The resulting attacks are predictable: PGD-discovered perturbations align with theoretically optimal perturbations derived from the interference geometry. Models trained on similar data develop similar interference patterns, explaining attack transferability. We then show that successful attacks on image classifiers exhibit the structure predicted by our proposed mechanism. These findings reveal that adversarial vulnerability can be a byproduct of networks' representational compression, complementing existing explanations based on data properties or architectural factors. Code for this paper is available at: github.com/Stevinson/adversarial-mechinterp

---

[1]Department of Computer Science, Imperial College London, London, United Kingdom. Correspondence to: Edward Stevinson <e.stevinson22@imperial.ac.uk>.

*Proceedings of the $43^{rd}$ International Conference on Machine Learning*, Seoul, South Korea. PMLR 306, 2026. Copyright 2026 by the author(s).

## 1. Introduction

Despite extensive study (Goodfellow et al., 2015; Eykholt et al., 2018; Bartoldson et al., 2024; Howe et al., 2025), adversarial examples remain poorly understood at a mechanistic level. They can be reliably generated, yet we lack a representation-level account that predicts the structure of successful perturbations and explains why attacks transfer between models. Such an account could inform more principled defences by linking robustness interventions to representation structure. Existing explanations attributing vulnerability to decision boundary imperfections (Moosavi-Dezfooli et al., 2016), properties of high dimensionality (Gilmer et al., 2018), or to non-robust patterns in training data (Ilyas et al., 2019) provide valuable insights but limited quantitative predictions about attack characteristics.

We approach this gap through the lens of *superposition*, a phenomenon in which neural networks represent more concepts, termed *features*, than they have dimensions by encoding them as non-orthogonal directions in activation space (Elhage et al., 2022). This packing increases representational capacity at the cost of introducing interference: activating one feature affects others in non-obvious ways. We demonstrate that gradient-based adversarial attacks can exploit this interference. Because features share representational dimensions, perturbations can simultaneously amplify a target class while suppressing the source through a combination of constructive and destructive interference.

Our analysis reveals a mechanistic pathway from data statistics to adversarial vulnerability: 1) input correlations constrain how features arrange in superposition, 2) these arrangements determine interference patterns, and 3) interference patterns dictate which perturbations succeed and why they transfer between models. This account yields testable predictions: attacks should align with optimal perturbations derived from the geometry, resulting in attack transferability when models have similar representations.

We validate these predictions in controlled settings where superposition can be precisely manipulated via dimensional bottlenecks. In synthetic settings, gradient-discovered attacks align with theoretically optimal perturbations (cosine similarity $> 0.92$), and transfer rates increase from 5% to 98% as feature geometry converges across models. We then show that attacks on vision transformer (ViT) classifiers

with engineered bottlenecks exhibit the structure our theory predicts. Our main contributions are:

- A mechanistic account linking superposition geometry to adversarial vulnerability. We show how data correlations induce specific feature arrangements that directly determine realised perturbations, offering insight into attack transferability and class-specific susceptibility.

- Theoretical and empirical evidence that superposition is a *sufficient* (though not necessary) condition for adversarial vulnerability. Non-orthogonal representations create exploitable interference, complementing rather than replacing existing explanations.

- Validation on ViT classifiers on CIFAR-10/100, STL-10, and TinyImageNet, demonstrating that the geometric signatures predicted by our theory appear in realistic settings.

## 2. Background

We first review adversarial examples and the attack methods used throughout the paper. We then introduce definitions for the tools we use in our analysis, namely the linear representation hypothesis (LRH) and superposition. Formally, let $\mathbf{x} \in \mathcal{X}$ denote the input to a neural network (NN), $y \in \mathcal{Y}$ its label, and $\mathbf{h}^{(l)} \in \mathbb{R}^{d_l}$ the activation vector of the $l$-th layer.

**Adversarial attacks**. Adversarial examples are inputs modified by perturbations that cause misclassification despite not changing the true class, *i.e.* a human would still classify the perturbed input correctly. We define the notion of adversarial perturbations and attacks following (Szegedy et al., 2014; Costa et al., 2024):

**Definition 1** (Adversarial Example). *Let $f : \mathcal{X} \rightarrow \mathcal{Y}$ be a classifier and $\mathbb{B}_p(\mathbf{x}, \varepsilon) = \{\mathbf{x}' : \|\mathbf{x}' - \mathbf{x}\|_p \leq \varepsilon\}$ the $\varepsilon$-ball around $\mathbf{x}$ in the $\ell_p$ norm. An input $\mathbf{x}' = \mathbf{x} + \boldsymbol{\delta}$ with $\mathbf{x}' \in \mathbb{B}_p(\mathbf{x}, \varepsilon)$ is an adversarial example if $f(\mathbf{x}') \neq f(\mathbf{x})$ (untargeted) or $f(\mathbf{x}') = y_{\text{target}}$ (targeted).*

We generate adversarial examples using projected gradient descent (PGD) (Madry et al., 2018), which iteratively follows the gradient of the classification loss $\mathcal{L}$ while projecting onto the constraint set. For $\ell_\infty$ attacks:

$$\mathbf{x}^{(t+1)} = \Pi_{\mathbb{B}_\infty(\mathbf{x}, \varepsilon)} \left( \mathbf{x}^{(t)} + \alpha \cdot \text{sign}\left( \nabla_{\mathbf{x}} \mathcal{L}(\mathbf{x}^{(t)}, y) \right) \right)$$

where $\Pi_{\mathbb{B}_\infty(\mathbf{x}, \varepsilon)}$ projects onto the $\ell_\infty$-ball of radius $\varepsilon$ around $\mathbf{x}$, and $\alpha$ is the step size. PGD follows the direction of steepest loss increase, finding perturbations that maximise loss increase within the constraint set.

**Linear representation hypothesis (LRH)**. The LRH posits that NNs represent semantically meaningful properties of their inputs – concepts like "is in French" – as linear directions in activation space (Park et al., 2024). These concepts

are denoted by a set of $M$ semantically meaningful *features*, $\mathcal{C} = \{c_1, \ldots, c_M\}$.

**Definition 2** (Linear Representation Hypothesis (LRH)). *A neural network layer with activations $\mathbf{h}^{(l)} \in \mathbb{R}^{d_l}$ satisfies the LRH if it represents the latent features $\mathcal{C} = \{c_1, \ldots, c_M\}$ as feature vectors $\{\mathbf{v}_j\}_{j=1}^M \subset \mathbb{R}^{d_l}$ such that:*

$$\mathbf{h}^{(l)}(\mathbf{x}) \approx \sum_{j=1}^M a_j(\mathbf{x})\, \mathbf{v}_j$$

*where $a_j(\mathbf{x}) \in \mathbb{R}$ represents the activation magnitude of feature $c_j$, and $\mathbf{v}_j$ is the corresponding direction. The features are **linearly accessible**: inputs $\mathbf{x}_0, \mathbf{x}_1 \in \mathcal{X}$ that differ mainly in the value of feature $c_j$ while holding others approximately fixed (i.e., $a_j(\mathbf{x}_1) - a_j(\mathbf{x}_0) = k$ and $|a_i(\mathbf{x}_1) - a_i(\mathbf{x}_0)| < \lambda$ for all $i \neq j$ and some small $\lambda > 0$) satisfy:*

$$\mathbf{h}^{(l)}(\mathbf{x}_1) - \mathbf{h}^{(l)}(\mathbf{x}_0) \approx k\mathbf{v}_j$$

*where $k$ reflects the change in $c_j$.*

Substantial evidence supports the LRH. Features encoding language (Gurnee et al., 2023), board game states (Nanda et al., 2023), and geographic locations (Templeton et al., 2024) have been identified via linear probes (Alain & Bengio, 2017) and sparse autoencoders (SAEs) (Bricken et al., 2023). Crucially, intervening on these directions causally changes model outputs, confirming they are used in computation rather than merely correlated with it.

**Superposition**. NNs are capable of representing far more features than there are available dimensions in activation space: $M \gg d_l$. For example, language models can reference many more place names than they have residual stream dimensions. This is possible through superposition: encoding concepts as non-orthogonal directions $\{\mathbf{v}_j\}_{j=1}^M$. Formally:

**Definition 3** (Superposition). *A set of $M$ features with representations $\{\mathbf{v}_j\}_{j=1}^M \subset \mathbb{R}^d$ is in superposition if:*

1. *(**Interference**) $M > d$ and there exist $i \neq j$ such that $\langle \mathbf{v}_i, \mathbf{v}_j \rangle \neq 0$.*
2. *(**Recoverability**) There exists a decoder $\psi : \mathbb{R}^d \rightarrow \mathbb{R}^M$ s.t. feature activations can be approximately recovered: for $\mathbf{h} = \sum_j a_j \mathbf{v}_j$, we have $\psi(\mathbf{h})_j \approx a_j$ for all $j$.*

The network trades representational capacity against feature interference by packing more features than dimensions. Condition 1 captures the cost of superposition: features interfere with each other, *i.e.* activating one feature activates others. Condition 2 ensures features remain usable despite interference. Whether networks mitigate this interference through non-linear operations (*e.g.* ReLU, softmax) (Gurnee et al., 2023) or use it constructively (Prieto et al., 2026) is an open research question.

**Interference**. Suppose the feature vectors $\{\mathbf{v}_i\}_{i=1}^M$ are in superposition, so activating one feature partially activates the others. The readout of feature $j$ is the projection of the representation $\mathbf{h}^{(l)} = \sum_{i=1}^M a_i \mathbf{v}_i$ onto its direction $\mathbf{v}_j$:

$$\mathbf{v}_j^\top \mathbf{h}^{(l)} = a_j + \sum_{i \neq j} a_i \left( \mathbf{v}_j^\top \mathbf{v}_i \right).$$

The first term is feature $j$'s own activation, while the second is interference from co-active features, weighted by their non-orthogonality with $\mathbf{v}_j$. On natural inputs, sparsity keeps this interference manageable (most $a_i = 0$). However, an adversary can craft inputs that activate features in combinations natural data never produces.

**Prevalence of superposition**. The condition $M > d$ is common in modern architectures. A clear illustration is ImageNet classifiers, which read 1,000 class logits from representations of a few hundred dimensions (*e.g.* ResNet-18: $d = 512$). Similarly, the unembedding matrix of an LLM maps a residual stream of dimension $d$ to a vocabulary of size $|\mathcal{V}| \gg d$ (*e.g.* Llama 3 8B: $d = 4096$, $|\mathcal{V}| > 128{,}000$). The phenomenon is also pervasive in intermediate representations such as MLPs (Elhage et al., 2022; Bricken et al., 2023) and attention, where per-head QK and OV circuits route the residual stream through subspaces of dimension $d_{\text{head}}$, with recent work providing evidence that features are superposed across and within heads (He et al., 2026; Jermyn et al., 2025). Superposition is therefore the norm in widely deployed architectures.

## 3. Setup

### 3.1. Studying superposition via classification heads

To study how adversarial attacks exploit superposition, we require a setting where feature vectors are known and interference is directly measurable. Classification heads provide a natural setting. For a classification head with weight matrix $\mathbf{W} \in \mathbb{R}^{k \times m}$, the logit for class $c$ is $z_c = \mathbf{w}_c^\top \mathbf{h}$, where $\mathbf{w}_c$ is row $c$ of $W$. These rows serve as *class feature vectors* – the directions along which the network reads evidence for each class. When the number of classes exceeds the representation dimension ($k > m$), or when semantically similar classes share representational structure, class vectors become non-orthogonal and interfere.

This framing offers key advantages: feature vectors can be read directly from $W$ without requiring discovery methods (e.g., SAEs, probes) that introduce approximation error; ground-truth feature labels of the respective classes are known; and interference between classes is directly measurable via $\mathbf{w}_i^\top \mathbf{w}_j$. By inserting a dimensional bottleneck before the classification head, we can systematically control the degree of superposition.

We study classification heads in three settings:

- **Synthetic argmax task:** A minimal bottleneck network with a closed-form input-to-class mapping. This permits computation of a ground truth after perturbation, control over superposition geometry, and direct comparison between empirical attacks and theoretical optima.

- **Classification with representation bottleneck:** ViTs trained on image datasets (CIFAR-10/100, STL-10, Tiny-ImageNet). This tests whether the geometric signatures predicted by our theory appear in practical settings with multiple layers of nonlinearities.

- **Classification without bottleneck:** Even without a bottleneck layer, the reduction in representation rank under regularised training is sufficient to invoke the superposition regime and associated vulnerability.

We develop our theoretical framework and core findings in the synthetic setting (Sec. 4), then validate that the predicted structure emerges in realistic networks (Sec. 5).

**Synthetic setting: Argmax task**. The synthetic task is constructed to: (1) represent class concepts as linear directions per the LRH; (2) induce controlled superposition via a dimensional bottleneck; (3) maintain a known mapping from inputs to the superposed features; and (4) permit exact computation of the true class after any perturbation, enabling testable predictions about adversarial mechanisms.

Let $\mathbf{x} \in \mathbb{R}^d$ with $d = kp$, partitioned as $\mathbf{x} = [\mathbf{x}^{(1)}, \ldots, \mathbf{x}^{(k)}]$, where each block $\mathbf{x}^{(j)} \in \mathbb{R}^p$ holds $p$ evidence components for class $j$. Following Elhage et al. (2022), we sample $\mathbf{x}$ from a sparse distribution $\mathcal{D}_\mathbf{x}$ over $[0,1]^d$. The correlation structure of $\mathcal{D}_\mathbf{x}$ is varied across experiments (see Sec. 4.2); in the uncorrelated case, each component is independently zero with probability $S$ (i.e., $\mathbb{P}(x_i^{(j)} = 0) = S$) and otherwise drawn $x_i^{(j)} \sim \text{Uniform}(0,1)$. The classification target is $y = \text{argmax}_{j \in [k]} \sum_{i=1}^p x_i^{(j)}$: each $x_i^{(j)}$ represents evidence for class $j$, and the label is the class with greatest total evidence.

We consider a two-layer linear bottleneck network with hidden dimension $m < k < d$:

$$\mathbf{h} = \mathbf{W}_e \, \mathbf{x} \in \mathbb{R}^m, \qquad \mathbf{z} = \mathbf{W}_d \, \mathbf{h} \in \mathbb{R}^k$$

trained with cross-entropy loss. The bottleneck forces superposition: the network must compress $d = kp$ input dimensions into $m < k$ latent dimensions and then recover sufficient information to identify the class with the largest sum, inducing non-orthogonal class representations. The bottleneck is designed to mimic a residual stream in a transformer-like model, often thought of as a store of features in superposition that does not itself perform computation (Elhage et al., 2021; Hänni et al., 2024). To verify generality, we also test a nonlinear variant with ReLU activations and biases. Models achieve $> 95\%$ accuracy.

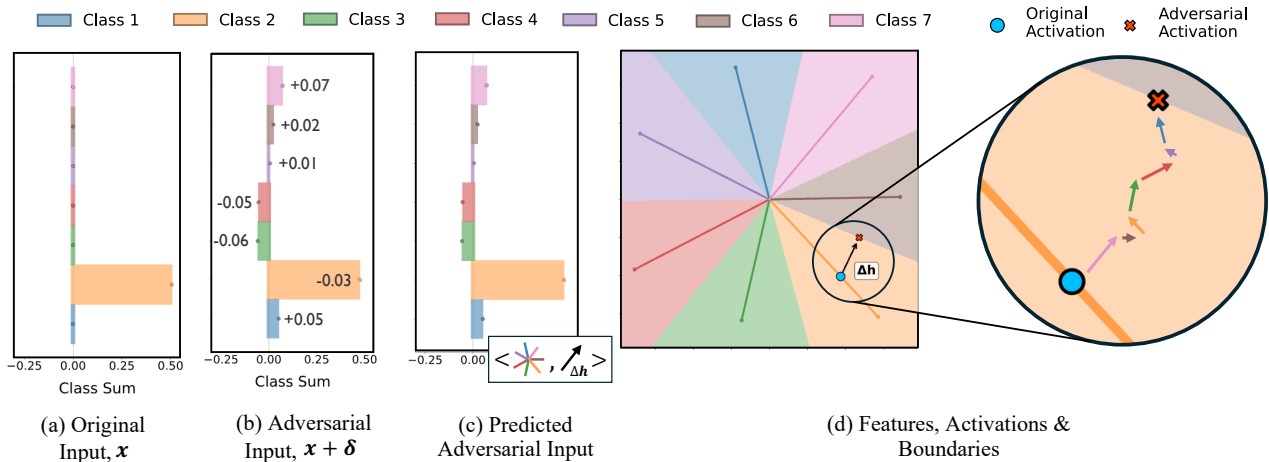

*Figure 1.* **An adversarial attack exploiting superposition geometry** ($k = 7$, $m = 2$, $p = 1$). **(a)** The original sample. **(b)** The adversarially perturbed sample, whose ground truth remains the same but is misclassified. The sign and magnitude of input perturbations are determined by the configuration of feature vectors. **(c)** The predicted optimal attack, matching the PGD-discovered adversary. **(d)** The original and adversarial sample in activation space. The coloured arrows show the effective class representations in the bottleneck. The attack shifts the representation along $\Delta \mathbf{h}$, with each input perturbation signed and scaled by its class direction's alignment with this shift.

We denote the columns of $\mathbf{W}_e$ by $\mathbf{v}_{c,r}$, where $c$ indexes the class and $r \in [p]$ indexes evidence components. Since each input coordinate contributes evidence to a single class, the encoder learns approximately shared directions within each class block, $\mathbf{v}_{c,r} \approx \mathbf{v}_c$. The decoder row $\mathbf{w}_c^\top$ of $\mathbf{W}_d$ reads out class-$c$ evidence and, empirically in this setting, aligns with this class-level encoder direction, $\mathbf{w}_c \approx \mathbf{v}_c$.

**Realistic Setting: ViT on CIFAR**. Our primary architecture is a ViT (Dosovitskiy et al., 2020) (12 layers, patch size $4 \times 4$, 6 heads, embedding dimension 384, MLP dimension 1536) trained from scratch on CIFAR-10/100 (Krizhevsky, 2009), STL-10 (Coates et al., 2011), and TinyImageNet (Tiny-IN) (Le & Yang, 2015). A bottleneck layer is inserted before the classification head, with bottleneck ratio $m/k \in \{0.2, 0.3, 0.5, 1.0\}$ controlling superposition pressure (e.g. $m \in \{20, 30, 50, 100\}$ for CIFAR-100's 100 classes). We additionally study a non-bottleneck variant (Sec. 5.3), where weight decay alone induces an effective low-rank bottleneck.

For each configuration, we train 5 models with different random seeds. Unlike the synthetic setting where we have a closed-form mapping from inputs to classes, here we focus on whether the structure of adversarial perturbations reflects the interference geometry of the classification head. Full training details are provided in Sec. D.1.

### 3.2. Evaluation metrics

In the synthetic setting, we define a successful adversarial example as satisfying: (1) the predicted class changes under perturbation, and (2) the true class (recomputed via group sums) remains unchanged. In the realistic setting, we use the standard definition where the predicted class changes

while the original label is preserved. Attacks are generated via $\ell_\infty$- and $\ell_2$-norm PGD with step size $\epsilon/4$, 50 steps, and 5 restarts. For successful adversarial examples, $\mathbf{x}' = \mathbf{x} + \boldsymbol{\delta}$, we measure:

1. **Input perturbation profile (IPP):** The sign and magnitude of each $\delta_i$.
2. **Latent attack alignment:** The similarity $\Delta \mathbf{h} \cdot \mathbf{w}_c$ between the latent attack vector $\Delta \mathbf{h} = \mathbf{h}_{\text{adv}} - \mathbf{h}_{\text{orig}}$ and class $c$'s representation.
3. **Attack transferability:** Success rate of attacks generated on one model when applied to another.
4. **Robust accuracy:** Fraction of examples correctly classified after perturbation with $\|\boldsymbol{\delta}\|_p \leq \epsilon$.

## 4. Superposition geometry determines adversarial attacks

We investigate if and how adversarial attacks exploit the interference inherent in superposed representations. Specifically, we address three questions:

> **1**. *Do adversarial perturbations exploit interference between superposed features?*
> **2**. *Do input correlations shape the geometric arrangement of latent features?*
> **3**. *Can shared geometry explain why attacks transfer between models?*

### 4.1. Do attacks exploit interference between superposed features?

An attack must move a sample across a decision boundary to change its class, but what determines the required input

perturbations? To answer this, we first characterise the optimal attack strategy in our linear setting.

**Proposition 1.** *The optimal input perturbation $\delta$ that maximally increases the pairwise margin $z_k - z_j$ under constraint $\|\delta\|_2 \leq \epsilon$ satisfies*

$$\delta \propto \mathbf{W}_e^\top \mathbf{n}_{jk}, \qquad \mathbf{n}_{jk} = \mathbf{w}_k - \mathbf{w}_j,$$

*where $\mathbf{n}_{jk}$ is the normal to the pairwise decision boundary between classes $j$ and $k$.*

*Proof sketch.* To increase the pairwise margin from class $j$ to class $k$, we maximise $z_k' - z_j' = \mathbf{n}_{jk}^\top(\mathbf{h} + \Delta\mathbf{h})$ where $\Delta\mathbf{h} = \mathbf{W}_e\delta$. The term $\mathbf{n}_{jk}^\top\mathbf{h}$ is independent of $\delta$, so this reduces to $\max_{\|\delta\|_2 \leq \epsilon} \delta^\top \mathbf{W}_e^\top \mathbf{n}_{jk}$. By Cauchy–Schwarz, this is maximised when $\delta \propto \mathbf{W}_e^\top \mathbf{n}_{jk}$. Proof in App. B.

**Corollary 1** (Interference drives vulnerability). *The adversarial perturbation magnitude for coordinate $r$ in class $i$ is $|\delta_{i,r}| \propto |\mathbf{w}_i^\top(\mathbf{w}_k - \mathbf{w}_j)|$, directly proportional to the interference between class feature vector $i$ and the class representations.*

This result reveals how superposition creates adversarial vulnerability: each input feature is perturbed proportionally to its geometric relationship with the class decision boundary. Under superposition, non-orthogonal representations cause semantically unrelated features to interfere with the class decision; adversarial perturbations exploit these dependencies to manipulate outputs.

**Empirical validation**. We test whether PGD-discovered attacks match these theoretical predictions. Fig. 1 shows a typical adversarial example: an input of Class 2 (orange) and its perturbed value that misclassifies it as Class 6 (brown). The perturbations appear arbitrary but follow a precise pattern mediated by latent geometry. The sign and magnitude of $\delta^{(j)}$ correlate strongly with how their class's feature vector $\mathbf{v}^{(j)}$ aligns with the latent attack vector $\Delta\mathbf{h}$. Positively aligned representations ($\mathbf{v}^{(j)} \cdot \Delta\mathbf{h} > 0$) see proportionally amplified inputs, while negatively aligned representations experience attenuation. Fig. 1(d) illustrates how these move the sample across the decision boundary.

For each configuration $(k, m)$ we train 5 models with different random seeds and run $\ell_2$-PGD attacks. For each

*Table 1.* Alignment between PGD attacks and theoretically optimal perturbations (cosine similarity, mean $\pm$ std dev across 5 seeds). Attacks are $\ell_2$-PGD at $\epsilon = 0.2$. Paired $t$-tests against the random baseline give $p < 10^{-5}$ for every row.

| $k$ | $m$ | Sim. (PGD vs Theory) | Sim. (Random) |
|-----|-----|----------------------|---------------|
| 6 | 2 | $0.97 \pm 0.02$ | $0.00 \pm 0.01$ |
| 30 | 10 | $0.96 \pm 0.00$ | $0.00 \pm 0.01$ |
| 90 | 30 | $0.92 \pm 0.00$ | $0.00 \pm 0.00$ |

successful adversarial example we compute the cosine similarity between the PGD perturbation $\delta$ and the theoretical optimum from Prop. 1. To account for the dependence between attacks generated from a single trained model, we conservatively compute the mean similarity *per seed* and then test across seeds. As reported in Tab. 1, PGD aligns strongly with the theoretical direction across configurations, and paired $t$-tests against a random-direction baseline are highly significant ($p < 10^{-5}$). As these theoretical directions leverage interference, we conclude that PGD systematically exploits superposition geometry.

> Adversarial attacks systematically exploit interference between superposed features. Successful PGD attacks are predictable given the specific superposition geometry.

### 4.2. Do input correlations determine latent feature geometry?

We next investigate whether correlations in the input data determine the geometric arrangements of feature vectors, and the mechanism that underlies this relationship. We hypothesise *interference avoidance*: frequently co-activating features arrange to minimise mutual interference. Stronger correlations impose tighter constraints on the learned geometry. Uncorrelated data permits any class-separating arrangement, while strong correlations determine a unique geometry up to rotation.

**Empirical validation**. We test this hypothesis by introducing three correlation conditions in the training data, and test how these constrain the degrees of freedom in learned representations.

- *Uncorrelated*: input dimensions sampled independently.
- *Paired*: each input in class $i$ is coupled with an input in class $j$, such that when one activates, so does its pair.
- *Global*: for each sample we draw a random phase $\phi$, and the probability that an input for class k is active is $p_k = p_{base} + A\cos(\phi + 2\pi k/K)$, meaning adjacent classes in index frequently co-activate.

We quantify geometric similarity by comparing pairwise cosine similarity matrices between feature vectors across models trained with different random seeds. Fig. 2 displays the arrangement of $\mathbf{v}^{(j)}$ for $m = 2$, $k = 7$.

For each correlation condition, we tested 45 seed pairs per condition. The results show a monotonic relationship: uncorrelated data yields highly variable geometries across seeds, paired correlations partially constrain the arrangements, whilst global correlations force near-identical geometries. Pairwise two-sample $t$-tests with Bonferroni correction confirm statistical significance ($p < 10^{-3}$ for all comparisons).

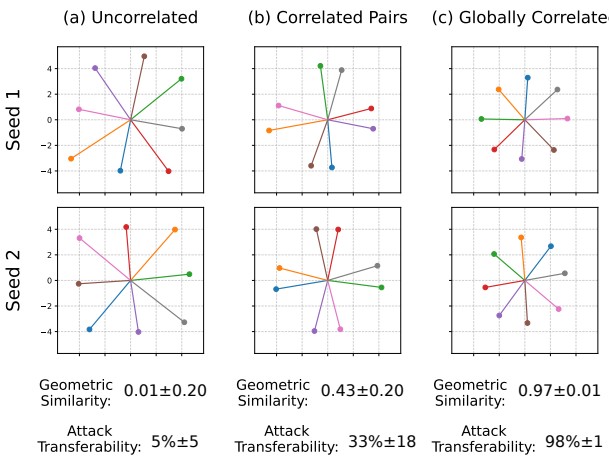

*Figure 2.* **Input correlations constrain representation geometry, which governs transferability**. Uncorrelated features are arranged at arbitrary angles across seeds; paired correlations constrain paired features to be orthogonal but leave the rest free; global correlations fix the cyclic ordering. Attack transfer scales with this constraint, from 5% to 98%.

Input correlations constrain feature geometry. Stronger correlations reduce geometric degrees of freedom, forcing different initialisations toward similar arrangements.

### 4.3. Does shared geometry explain attack transferability?

Attacks tailored for specific interference patterns should transfer between models with similar feature arrangements.

**Proposition 2.** *Models with feature representations related by orthogonal transformation $\mathbf{Q}$ (where $\mathbf{Q}^\top \mathbf{Q} = \mathbf{I}$) share identical optimal attack directions in input space.*

*Proof sketch.* Optimal perturbations are invariant under orthogonal transformation: $\delta_i' \propto (\mathbf{v}_i')^\top (\mathbf{v}_k' - \mathbf{v}_j') = \mathbf{v}_i^\top \mathbf{Q}^\top \mathbf{Q}(\mathbf{v}_k - \mathbf{v}_j) = \mathbf{v}_i^\top (\mathbf{v}_k - \mathbf{v}_j) \propto \delta_i$. Proof in App. B.

**Empirical validation**. Using the same seed pairs as the previous analysis, we generate attacks on source models and evaluate transfer success. Transfer rates correlate strongly with geometric similarity (Fig. 2): global correlations yield $98\% \pm 11\%$ transfer, paired correlations produce $33\% \pm 18\%$, and uncorrelated data shows only $5\% \pm 5\%$.

Each perturbation creates constructive interference that pushes representations across decision boundaries. When applied to a model with different geometry, interference patterns that were previously constructive become destructive, causing the attack to fail.

Attack transferability is governed by shared interference patterns. Models with similar superposition geometry exhibit high transfer rates.

### 4.4. Does reducing superposition suppress attacks?

If superposition creates vulnerability through interference, removing it should eliminate adversarial examples. We test this prediction with three experiments.

**Non-compressed bottleneck**. When $m = k$ the network learns to represent each $\mathbf{v}^{(j)}$ orthogonally. Any perturbation that changes the model's prediction also changes the ground truth class, *e.g.* moving a sample from class A to class B requires making the sum of class B features exceed class A's – genuinely transforming it into a class B sample. We observe zero successful adversarial examples across 1000 attempts at all tested $\epsilon$ values.

**Isolating superposition from capacity**. When fixing $m$ and varying $k$, robust accuracy decreases monotonically with $k/m$ (Tab. 7), demonstrating that vulnerability scales with the degree of superposition.

**Isolating superposed features**. Constraining one class vector $\mathbf{v}_\perp$ to remain orthogonal to all others during training causes attacks between classes that remain in superposition to leave the orthogonal class's inputs unperturbed.

Adversarial attacks use their budget to exploit those features in superposition.

Our theoretical and empirical results establish a **mechanistic pathway**. Specifically: (i) input correlations constrain feature arrangements in superposition, (ii) these geometric arrangements determine interference patterns, (iii) interference patterns dictate optimal perturbations via $\delta \propto \mathbf{W}_e^\top (\mathbf{w}_k - \mathbf{w}_j)$ (Prop. 1), and (iv) shared geometric constraints yield similar interference patterns across models, enabling transferability (Prop. 2):

**Correlations** $\xrightarrow{\text{constrain}}$ **Feature Geometry** $\xrightarrow{\text{determines}}$
**Interference Patterns** $\xrightarrow{\text{enable}}$ **Transferability**

## 5. Attacks on image classifiers

We now test whether the geometric account established in our synthetic setting extends to image classifiers. Our theory makes three predictions:

1. Interference between class feature vectors should predict attack structure in input space.
2. This relationship should strengthen as bottleneck dimension decreases, increasing superposition pressure.
3. Interference patterns should be consistent across random seeds, reflecting structure rather than optimisation noise.

Unlike the synthetic setting, image classifiers involve multiple layers and nonlinearities mediating the relationship be-

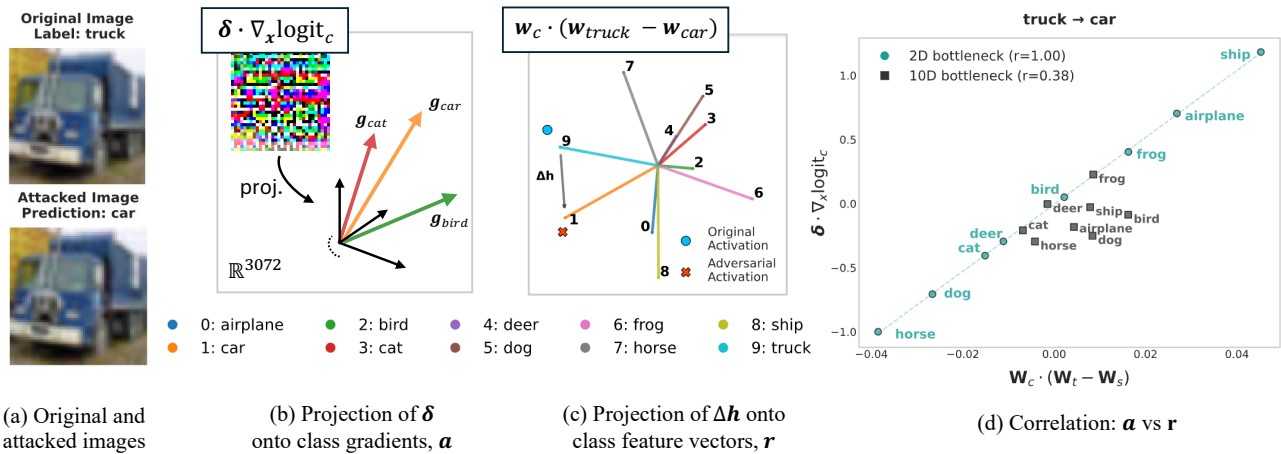

(a) Original and attacked images | (b) Projection of $\boldsymbol{\delta}$ onto class gradients, $\boldsymbol{a}$ | (c) Projection of $\Delta\boldsymbol{h}$ onto class feature vectors, $\boldsymbol{r}$ | (d) Correlation: $\boldsymbol{a}$ vs $\boldsymbol{r}$

*Figure 3.* **Attacks exploit superposition geometry in ViTs.** **(a)** An attack misclassifies a truck as a car. **(b)** The perturbation projected onto class attack gradients in input space. **(c)** The latent attack vector projected onto class feature vectors. **(d)** When the bottleneck is tight ($m = 2$), the latent profile $\mathbf{r}$ strongly predicts the input-space profile $\mathbf{a}$. As interference decreases ($m = 10$), the correlation diminishes.

tween pixel perturbations and representation-space effects. The key question is whether classification head geometry still governs attack structure despite this complexity.

We introduce our method to test these predictions under this increased model complexity (Sec. 5.1). We first examine models with a representation bottleneck ($m < k$; Sec. 5.2), then show that superposition-induced vulnerability can arise even without such a bottleneck, due to the tendency of NNs to learn low-rank representations (Sec. 5.3).

### 5.1. Measuring attack structure

To test whether latent interference predicts attack structure of the inputs, we define: (i) an input-space profile measuring how perturbations affect each class, (ii) a latent-space profile measuring geometric interference between class features.

**Input-space profile**. For each class $c$, define the *class attack gradient* $\mathbf{g}_c(\mathbf{x}) = \partial z_c / \partial \mathbf{x}$, the direction in input space that maximally increases the logit for class $c$. For a perturbation $\boldsymbol{\delta}$, the input-space attack profile is:

$$\mathbf{a}(\mathbf{x}, \boldsymbol{\delta}) = \left[\mathbf{g}_1(\mathbf{x})^\top \boldsymbol{\delta}, \ldots, \mathbf{g}_k(\mathbf{x})^\top \boldsymbol{\delta}\right]$$

This measures how much the perturbation pushes toward each class in pixel space.

**Latent-space profile**. For an attack from source class $s$ to target class $t$, the latent-space attack profile is:

$$\mathbf{r}_{s \to t} = \left[\mathbf{w}_1^\top (\mathbf{w}_t - \mathbf{w}_s), \ldots, \mathbf{w}_k^\top (\mathbf{w}_t - \mathbf{w}_s)\right]$$

This measures how each class feature vector aligns with the attack direction $\mathbf{w}_t - \mathbf{w}_s$ in the bottleneck, analogous to the latent attack alignment in Sec. 4. This profile depends only on the classification head weights and requires no input.

**Correlation metric**. For each successful attack, we compute the Pearson correlation between $\mathbf{a}$ and $\mathbf{r}$, excluding source and target classes. The latter are determined by the attack objective ($r_t > 0$, $r_s < 0$ by construction) and their inclusion would trivially inflate the correlation. We denote the mean correlation across successful attacks as $\bar{\rho}$.

**Interference metrics**. We compute the interference matrix $S_{ij} = \cos(\mathbf{w}_i, \mathbf{w}_j)$ and measure: (1) *interference magnitude* $I = {}^2/_{k(k-1)} \sum_{i<j} |S_{ij}|$, the mean absolute off-diagonal similarity; and (2) *interference consistency* $\kappa$, the Pearson correlation between interference matrices across seeds, with significance testing via $z$-scores against random Gaussian weights.

### 5.2. Results with bottleneck

**Interference predicts input-space attack structure**. Fig. 3 illustrates our core finding. To misclassify a truck as a car, the adversarial perturbation does not simply increase the car logit – it also perturbs seemingly unrelated classes (e.g. suppressing horse and amplifying ship), following a pattern predicted by the interference geometry in the classification head. This is not an isolated example: across attacks, the input-space profile $\mathbf{a}$ correlates strongly with the latent profile $\mathbf{r}$ (Tab. 2). This confirms Prediction 1: despite the complex nonlinear pathway from pixels to the classification head, interference geometry governs attack structure.

**Tighter bottlenecks amplify the effect**. As bottleneck dimension increases toward the number of classes, the correlation between $\mathbf{a}$ and $\mathbf{r}$ diminishes (Tab. 2). For CIFAR-10, $\bar{\rho}$ drops from 0.89 at $m = 2$ to 0.16 at $m = 10$; for CIFAR-100, from 0.61 at $m = 20$ to 0.34 at $m = 100$. With reduced interference, attacks align primarily with the source and target class directions, producing minimal collateral ac-

tivation of other classes. This mirrors our synthetic findings, where reducing superposition pressure narrowed the spread of attack effects. The same trend holds across perturbation budgets (App. Tab. 9). This confirms Prediction 2.

*Table 2.* **Correlation between latent and input-space attack profiles in ViTs** ($\varepsilon = 2/255$; additional in App. Tab. 9). $\bar{\rho}$: mean correlation between **r** and **a** (excluding source/target). $m/k = 0.2$ corresponds to $m = 2$ for CIFAR-10 and $m = 20$ for CIFAR-100. Mean $\pm$ std dev over 5 seeds.

| $m/k$ | 0.2 | 0.3 | 0.5 | 1.0 |
|---|---|---|---|---|
| CIFAR-10 | $0.89 \pm 0.01$ | $0.81 \pm 0.01$ | $0.59 \pm 0.02$ | $0.16 \pm 0.02$ |
| CIFAR-100 | $0.61 \pm 0.01$ | $0.48 \pm 0.01$ | $0.42 \pm 0.01$ | $0.34 \pm 0.01$ |
| STL-10 | $0.83 \pm 0.01$ | $0.74 \pm 0.02$ | $0.54 \pm 0.01$ | $0.18 \pm 0.02$ |
| Tiny-IN | $0.52 \pm 0.01$ | $0.45 \pm 0.01$ | $0.40 \pm 0.01$ | $0.33 \pm 0.01$ |

**Interference patterns are consistent across seeds**. The arrangement of class feature vectors in the bottleneck is not arbitrary. Models trained with different random seeds develop similar interference patterns (Tab. 3) with cross-seed consistency yielding $z$-scores far exceeding the significance threshold. This consistency suggests that superposition geometry is shaped by structure in the data – specifically, which classes share visual features – rather than by optimisation trajectories. Qualitatively, the learned arrangements reflect semantic similarity (*e.g.* 'cat' closer to 'dog'). Classes that share visual features develop non-orthogonal representations, creating predictable interference patterns that persist across training runs. This confirms Prediction 3.

> Adversarial attacks in vision transformers systematically exploit interference between superposed features, as predicted by our analysis in the synthetic setting. Classification head interference explains input-space attack structure ($\bar{\rho}$ up to 0.89), with effect size scaling with superposition pressure. Interference patterns are consistent across seeds and reflect data semantics.

### 5.3. Generalising results without a bottleneck

Previous sections used explicit bottlenecks ($m < k$) as a clean intervention to isolate superposition pressure. We now show our mechanism can help explain adversarial vulnerability even when $m \gg k$. NNs trained with gradient-based optimisation converge towards low-rank weight matrices and representations, especially under regularisation (Mousavi-Hosseini et al., 2022; Feng et al., 2022; Huh et al., 2023). This means in practice, whether increased interference occurs depends on the *intrinsic dimension* ($\bar{m}$), not the *ambient dimension* ($m$).

We train ViT models on CIFAR-10 under increasing weight decay, without explicit bottlenecks. We estimate $\bar{m}$ via PCA as the minimum principal components explaining $\tau =$

*Table 3.* **ViT performance and interference metrics.** *Acc.*: clean test accuracy; $I$: mean absolute interference; $\kappa$: cross-seed interference consistency. Mean $\pm$ std over 5 seeds. ResNet-50 replications in Sec. D.4 achieve higher and more bottleneck-stable clean accuracy while exhibiting the same monotonic trend in $I$, indicating these trends are not driven by accuracy loss at tighter bottlenecks.

| | $m/k$ | 0.2 | 0.3 | 0.5 | 1.0 |
|---|---|---|---|---|---|
| CIFAR-10 | Acc. | $58.7 \pm 1.2$ | $79.0 \pm 1.0$ | $87.4 \pm 0.5$ | $89.3 \pm 0.1$ |
| | $I$ | $0.61 \pm 0.1$ | $0.43 \pm 0.1$ | $0.23 \pm 0.1$ | $0.12 \pm 0.0$ |
| | $\kappa$ | 0.99 | 0.95 | 0.88 | 0.80 |
| CIFAR-100 | Acc. | $66.3 \pm 0.8$ | $66.2 \pm 0.8$ | $67.9 \pm 0.7$ | $70.2 \pm 0.8$ |
| | $I$ | $0.16 \pm 0.0$ | $0.12 \pm 0.0$ | $0.08 \pm 0.0$ | $0.06 \pm 0.0$ |
| | $\kappa$ | 0.89 | 0.90 | 0.89 | 0.92 |
| STL-10 | Acc. | $67.1 \pm 0.4$ | $82.8 \pm 0.5$ | $88.0 \pm 0.3$ | $89.0 \pm 0.2$ |
| | $I$ | $0.60 \pm 0.00$ | $0.46 \pm 0.01$ | $0.18 \pm 0.01$ | $0.12 \pm 0.00$ |
| | $\kappa$ | 0.99 | 0.96 | 0.96 | 0.95 |
| Tiny-IN | Acc. | $55.5 \pm 0.4$ | $56.5 \pm 0.4$ | $57.1 \pm 0.9$ | $58.8 \pm 0.4$ |
| | $I$ | $0.14 \pm 0.00$ | $0.10 \pm 0.00$ | $0.07 \pm 0.00$ | $0.06 \pm 0.00$ |
| | $\kappa$ | 0.90 | 0.95 | 0.96 | 0.99 |

$0.95$ of variance. Despite an embedding dimension of $384$, regularisation creates an effective bottleneck ($\bar{m} \ll 384$).

Results in Tab. 4 support our hypothesis: the predicted interference effects emerge as the effective dimension $\bar{m}$ decreases. Increasing weight decay decreases $\bar{m}$ and increases $\bar{\rho}$, mirroring the effect of explicit bottlenecks. This suggests that rank-reducing regularisation can amplify the interference-mediated vulnerability identified by our framework. This reveals that standard regularisation may inadvertently create adversarial vulnerabilities. These findings position superposition-induced vulnerability as a concern in realistic training scenarios and highlight representation dimensionality as a component of model safety.

*Table 4.* **Input class attack correlation across effective dimensions in the ViT for $\epsilon = 2/255$ on CIFAR-10**. Higher weight decay reduces effective dimension, producing similar effects to explicit bottlenecks. $\bar{\rho}$ is the mean correlation between latent attack profile **r** and input-space attack profile **a** (excluding source/target).

| $\lambda$ | 0.1 | 0.3 | 0.6 | 0.7 | 1.5 | 2.0 |
|---|---|---|---|---|---|---|
| $\bar{m}$ | 29 | 12 | 10 | 9 | 8 | 6 |
| $\bar{\rho}$ | 0.14 | 0.27 | 0.40 | 0.68 | 0.79 | 0.79 |

## 6. Discussion & related work

**Explanations for adversarial examples**. The field recognises that no single explanation accounts for all adversarial vulnerability, but rather a confluence of contributing factors. The linear hypothesis attributes vulnerability to model linearity, whereby small perturbations accumulate through layers

(Goodfellow et al., 2015). Our work refines this. Linearity enables gradient-based attacks, but vulnerability stems from *which* directions are sensitive, determined by superposition geometry. The non-robust features hypothesis (Ilyas et al., 2019) argues that models exploit predictive but brittle input patterns. In our account, robustness depends instead on how a feature's representation interacts with boundaries: the same feature could be robust if encoded orthogonally to decision-relevant directions, or brittle if it interferes.

Decision boundary analyses characterise vulnerability geometrically. Moosavi-Dezfooli et al. (2016) show boundaries have small mean curvature near data, allowing linear approximation, so minimal perturbations are approximately normal to the boundary. Shamir et al. (2021) study how non-linear boundary structure creates vulnerability. Our work complements these: non-orthogonal features determine boundary orientations, and the interference pattern specifies which normal directions are achievable with small perturbations. Work on manifold structure (Stutz et al., 2019) distinguishes on-manifold from off-manifold adversaries. Our framework connects to this: data correlations constrain feature geometry, and perturbations exploiting superposed features will generically be off-manifold as natural data does not activate features in the combinations that interference enables.

Attack transferability is incompletely understood. Prior work attributes limited transfer to representation discrepancies (Li et al., 2023; Wang et al., 2025), and shows that decorrelating features reduces transferability (Wiedeman & Wang, 2022). Our framework offers a common mechanism for these findings.

**Superposition and representation geometry**. Superposition research identifies feature interference mechanisms: Nanda (2022) distinguish *representational* and *computational* superposition; Gurnee et al. (2023) differentiate *alternating* and *simultaneous* interference. Data correlations shape these arrangements: correlated features become orthogonal (Elhage et al., 2022), drive superposition formation (Chan, 2024), and induce semantic clustering (Prieto et al., 2026). Gorton & Lewis (2025) investigate the link between superposition and adversarial robustness, showing that interventions on superposition can control vulnerability. Beyond superposition, representation geometry is shaped by spectral bias (Rahaman et al., 2019), neural collapse (Kothapalli, 2023), and optimisation objectives (Casper, 2023).

**Low-rank structure and adversarial vulnerability**. Superposition is tied to low-rank structure: encoding $k$ classes in $m < k$ dimensions yields a classification head $\mathbf{W} \in \mathbb{R}^{k \times m}$ of rank at most $m$. Prior work links low-rank weight matrices to adversarial vulnerability, attributing it to exploding singular values and poor conditioning (Barsbey et al., 2025; Savostianova et al., 2023). Our framework is consistent with this: the rank constraint forces class directions to be non-orthogonal, creating the interference that adversaries exploit. However, rank is necessary but not sufficient to determine an attack. Two heads of identical rank can induce entirely different interference patterns, and thus different perturbations. It is the specific feature geometry, not the compression ratio alone, that fixes which directions interfere. Our account refines the low-rank view by identifying *which* of the many rank-$m$ geometries a model realises as the quantity that governs its vulnerability.

**Perceptual alignment of gradients (PAG)**. Adversarially robust models develop input gradients aligned with semantically meaningful directions (Tsipras et al., 2019; Santurkar et al., 2019; Ganz et al., 2023; Srinivas et al., 2023). PAG is a property of *adversarially trained* models, concerning whether *input-space* gradients are perceptually meaningful. Our framework instead operates in activation space, showing that data correlations, low-rank structure, and weight decay enforce semantically meaningful feature geometry whose interference patterns are exploited by attacks.

## 7. Concluding remarks

We demonstrate that adversarial attacks can exploit interference patterns arising from the geometry of superposed features in NNs. Our experiments, spanning toy models and ViTs, establish that data properties - namely correlations and sparsity - induce distinct superposition geometries, which in turn create predictable adversarial vulnerabilities. We show that these geometric arrangements and the resulting interference allow for the prediction of phenomena such as attack transferability and class-specific susceptibility. Our new perspective frames adversarial vulnerability as a potential, inherent consequence of how networks efficiently encode vast amounts of information via superposition.

**Limitations & future work**. Our insights stem primarily from simplified models to isolate superposition's role, but the vast adversarial robustness literature suggests precise vulnerability mechanisms will likely vary by attack and model. Our experiments establish that superposition *suffices* for adversarial vulnerability, not that it is *necessary*. Attacks find whichever weakness requires the smallest perturbation, and superposition may dominate in capacity-constrained settings while being secondary elsewhere. We focus on features with known labels; extension to intermediate layers will require feature discovery via probes or dictionary learning (*e.g.* SAEs, preliminary results in App. E).

Future work includes quantifying the 'adversarial cost' of superposition, characterising how adversarial training reshapes feature geometry, and extending the analysis to intermediate layers. Understanding how superposition-induced vulnerability interacts with other explanatory factors will clarify when this mechanism dominates and when it is secondary.

## Acknowledgments

L. Prieto was supported by the UKRI Centre for Doctoral Training in Safe and Trusted AI [EP/S0233356/1]. M. Barsbey was supported by the EPSRC Project GNOMON (EP/X011364/1). T. Birdal acknowledges support from the Engineering and Physical Sciences Research Council [grant EP/X011364/1] and the UKRI Future Leaders Fellowship [grant number MR/Y018818/1].

## Impact statement

Due to its investigative nature, our framework is not anticipated to have negative impact or societal consequences. Our experiments use only publicly available datasets and synthetic toy models, with no human subjects or personally identifiable information. The gradient-based attack methods are well-established, and our work introduces no novel attacks but provides theoretical understanding of existing phenomena. By elucidating the relationship between superposition and adversarial vulnerability, we hope to inform research balancing model capability with robustness, contributing to trustworthy AI for critical applications. We declare no conflicts of interest.

## Reproducibility statement

Source code is available at https://github.com/Stevinson/adversarial-mechinterp. Sec. 4 includes architectural details, training procedures, evaluation metrics, and data generation processes for the synthetic model experiments. Sec. 5 provides architecture and training procedures for image model experiments, with further details in Sec. D.1. Experiments were conducted on an AMD Ryzen Threadripper 3970X 32-Core with 256GB RAM and RTX 3090. Additional SAE experiments are documented in Appendix App. E.

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

# Appendix

## A. Generation of adversarial examples

We use PGD to generate AEs, which is an iterative method to generate attacks (Madry et al., 2018). For untargeted attacks, it maximises the classifier's loss function $\mathcal{L}(f(\mathbf{x}'), y_{\text{true}})$; for targeted, it minimises $\mathcal{L}(f(\mathbf{x}'), y_{\text{target}})$ by the update rule:

$$\mathbf{x}'^{(k+1)} = \Pi_{\mathcal{S}} \left( \mathbf{x}'^{(k)} \pm \alpha \mathbf{g}_k \right) \tag{1}$$

($+$ for maximisation, $-$ for minimisation). $\mathbf{x}'^{(0)}$ is an initial perturbed input, $\alpha$ is step size, $\mathbf{g}_k$ is the (normalised) gradient $\nabla_{\mathbf{x}} \mathcal{L}(f(\mathbf{x}'^{(k)}), y_{\text{class}})$ ($y_{\text{class}}$ being $y_{\text{true}}$ or $y_{\text{target}}$), and $\Pi_{\mathcal{S}}$ projects onto the $\epsilon$-ball $\mathcal{S} = \{\mathbf{x}' \mid \|\mathbf{x}' - \mathbf{x}\|_p \le \epsilon\}$, with input domain clipping. We use the $\ell_p$ norms:

1. $\ell_\infty$: Constraint $\|\boldsymbol{\delta}\|_{\ell_\infty} = \max_i |\boldsymbol{\delta}_i| \le \epsilon$. $\mathbf{g}_k$ is $\text{sign}(\nabla_{\mathbf{x}} \mathcal{L}(\cdot))$. $\Pi_{\mathcal{S}}$ clips each $\mathbf{x}'_i$ to $[\mathbf{x}_i - \epsilon, \mathbf{x}_i + \epsilon]$.
2. $\ell_2$: Constraint $\|\boldsymbol{\delta}\|_{\ell_2} = \sqrt{\sum_i \delta_i^2} \le \epsilon$. $\mathbf{g}_k$ is $\nabla_{\mathbf{x}} \mathcal{L}(\cdot)/\|\nabla_{\mathbf{x}} \mathcal{L}(\cdot)\|_{\ell_2}$. $\Pi_{\mathcal{S}}$ rescales $\boldsymbol{\delta} = \mathbf{x}' - \mathbf{x}$ if $\|\boldsymbol{\delta}\|_{\ell_2} > \epsilon$ via $\boldsymbol{\delta} \leftarrow \epsilon \cdot \boldsymbol{\delta}/\|\boldsymbol{\delta}\|_{\ell_2}$.

## B. Omitted proofs & theoretical results

To understand how adversarial attacks exploit representation interference, we analyse how a perturbation in input space is pulled back from a vulnerable direction in the bottleneck representation. Conceptually, the vulnerability comes from latent-space interference. Class feature vectors are represented in a space too small for them to be mutually orthogonal, so moving along one direction changes projections onto others.

Recall the linear encoder-decoder model from Sec. 4: $\mathbf{h} = \mathbf{W}_e \mathbf{x} \in \mathbb{R}^m$, with $\mathbf{W}_e \in \mathbb{R}^{m \times d}$, and $\mathbf{z} = \mathbf{W}_d \mathbf{h} \in \mathbb{R}^k$, with $\mathbf{W}_d \in \mathbb{R}^{k \times m}$. We write the columns of $\mathbf{W}_e$ as $\{\mathbf{v}_i\}_{i=1}^d$, so $\mathbf{h} = \sum_i x_i \mathbf{v}_i$. We write $\mathbf{w}_c^\top$ for row $c$ of $\mathbf{W}_d$, so the class-$c$ logit is $z_c = \mathbf{w}_c^\top \mathbf{h}$. The decoder rows $\mathbf{w}_c$ are class feature vectors – they read class evidence from the bottleneck.

In the argmax task, the input has block structure $\mathbf{x} = [\mathbf{x}^{(1)}, \dots, \mathbf{x}^{(k)}]$, with $\mathbf{x}^{(c)} \in \mathbb{R}^p$ and $d = kp$. The target depends on the class evidence sums $s_c = \sum_{r=1}^p x_r^{(c)}$. Writing the encoder column for coordinate $r$ in class block $c$ as $\mathbf{v}_{c,r}$, we empirically observe that within-block columns learn approximately shared directions, $\mathbf{v}_{c,r} \approx \mathbf{u}_c$, where $\mathbf{u}_c$ is the effective class direction. The decoder row $\mathbf{w}_c^\top$ reads out class-$c$ evidence and, empirically, aligns with the same direction: $\mathbf{w}_c \approx \mathbf{u}_c$. Thus our synthetic model is close to a symmetric superposition setting (Elhage et al., 2022) in that class evidence is written into and read from the bottleneck along the same effective directions.

The pairwise decision boundary between classes $j$ and $k$ is

$$\mathcal{B}_{jk} = \{\mathbf{h} : z_j = z_k\} = \{\mathbf{h} : (\mathbf{w}_k - \mathbf{w}_j)^\top \mathbf{h} = 0\}.$$

The vector $\mathbf{n}_{jk} = \mathbf{w}_k - \mathbf{w}_j$ is the normal to this pairwise boundary, pointing in the direction that increases the margin $z_k - z_j$.

**Representation-level interference**. We first note the latent-space mechanism. If a perturbation moves the representation by $\Delta \mathbf{h} \propto \mathbf{w}_k - \mathbf{w}_j$, then the change in the logit of any class $\ell$ is

$$\Delta z_\ell = \mathbf{w}_\ell^\top \Delta \mathbf{h} \propto \mathbf{w}_\ell^\top (\mathbf{w}_k - \mathbf{w}_j).$$

Thus the non-target classes affected by an attack, and the signs and magnitudes of those effects, are determined by the Gram structure of the class readout directions. In the symmetric setting where $\mathbf{w}_c \approx \mathbf{u}_c$, this is interference between latent class feature directions:

$$\Delta z_\ell \propto \mathbf{u}_\ell^\top (\mathbf{u}_k - \mathbf{u}_j).$$

This is the sense in which superposition creates vulnerability: because the class directions are non-orthogonal, moving toward one class necessarily changes projections onto others.

**Proposition 1 (Optimal pairwise margin perturbation)**. The optimal input perturbation $\boldsymbol{\delta}$ that maximally increases the pairwise margin $z_k - z_j$ under the constraint $\|\boldsymbol{\delta}\|_2 \le \epsilon$ satisfies

$$\boldsymbol{\delta} \propto \mathbf{W}_e^\top \mathbf{n}_{jk}, \qquad \mathbf{n}_{jk} = \mathbf{w}_k - \mathbf{w}_j,$$

where $\mathbf{n}_{jk}$ is the normal to the pairwise decision boundary between classes $j$ and $k$.

*Proof.* An input perturbation $\boldsymbol{\delta}$ induces latent perturbation $\Delta\mathbf{h} = \mathbf{W}_e\boldsymbol{\delta}$. Under this perturbation, the pairwise margin becomes

$$z'_k - z'_j = \mathbf{w}_k^\top(\mathbf{h} + \Delta\mathbf{h}) - \mathbf{w}_j^\top(\mathbf{h} + \Delta\mathbf{h}) \tag{2}$$

$$= (\mathbf{w}_k - \mathbf{w}_j)^\top\mathbf{h} + (\mathbf{w}_k - \mathbf{w}_j)^\top\mathbf{W}_e\boldsymbol{\delta}. \tag{3}$$

The first term is independent of $\boldsymbol{\delta}$, so maximising the change in margin is equivalent to solving

$$\max_{\|\boldsymbol{\delta}\|_2 \leq \epsilon} \boldsymbol{\delta}^\top\mathbf{W}_e^\top(\mathbf{w}_k - \mathbf{w}_j).$$

Let $\mathbf{g}_{jk} = \mathbf{W}_e^\top(\mathbf{w}_k - \mathbf{w}_j)$. By Cauchy-Schwarz,

$$\boldsymbol{\delta}^\top\mathbf{g}_{jk} \leq \|\boldsymbol{\delta}\|_2\|\mathbf{g}_{jk}\|_2 \leq \epsilon\|\mathbf{g}_{jk}\|_2.$$

Equality is achieved when $\boldsymbol{\delta}$ is parallel to $\mathbf{g}_{jk}$, giving

$$\boldsymbol{\delta}^\star = \epsilon\frac{\mathbf{W}_e^\top(\mathbf{w}_k - \mathbf{w}_j)}{\|\mathbf{W}_e^\top(\mathbf{w}_k - \mathbf{w}_j)\|_2},$$

$\square$

Intuitively, the attack identifies the vulnerable latent direction that increases the target readout relative to the source readout. The encoder transpose maps this latent direction back into input space. Thus superposition determines the vulnerable latent direction, while the encoder determines which input perturbation realises it.

**Corollary 1 (Interference drives vulnerability)..** For input coordinate $i$, the adversarial perturbation magnitude satisfies

$$|\delta_i| \propto \left|\mathbf{v}_i^\top(\mathbf{w}_k - \mathbf{w}_j)\right|,$$

where $\mathbf{v}_i$ is the encoder direction for input coordinate $i$.

For coordinate $r$ in class block $c$, this gives $|\delta_{c,r}| \propto \left|\mathbf{v}_{c,r}^\top(\mathbf{w}_k - \mathbf{w}_j)\right|$. Since $\mathbf{v}_{c,r} \approx \mathbf{u}_c$ and $\mathbf{u}_c \approx \mathbf{w}_c$ in our synthetic setting, this is equivalently

$$|\delta_{c,r}| \propto \left|\mathbf{w}_c^\top(\mathbf{w}_k - \mathbf{w}_j)\right|.$$

Thus, in the synthetic setting, the input perturbation profile is determined by interference between class feature vectors. Coordinates whose class readout aligns with $\mathbf{w}_k - \mathbf{w}_j$ are increased; coordinates whose class readout anti-aligns are suppressed; and the magnitude scales with the strength of this alignment.

**Remark on multiclass targeted attacks..** Proposition 1 characterises the perturbation that maximally increases the pairwise margin $z_k - z_j$. In a multiclass classifier, a targeted adversarial example for class $k$ must also make $z_k$ exceed all other logits. Therefore, if another class lies between $j$ and $k$, a successful targeted attack may be governed by additional pairwise boundaries.

**Proposition 2 (Shared geometry implies shared attacks)**. Models with encoder directions and class readout vectors related by the same orthogonal transformation $\mathbf{Q}$, where $\mathbf{Q}^\top\mathbf{Q} = \mathbf{I}$, share identical optimal attack directions in input space.

*Proof.* Suppose two models have

$$\mathbf{W}'_e = \mathbf{Q}\mathbf{W}_e, \qquad \mathbf{w}'_c = \mathbf{Q}\mathbf{w}_c \quad \text{for all classes } c.$$

Then the optimal attack direction for the second model is

$$(\mathbf{W}'_e)^\top(\mathbf{w}'_k - \mathbf{w}'_j) = (\mathbf{Q}\mathbf{W}_e)^\top\mathbf{Q}(\mathbf{w}_k - \mathbf{w}_j) \tag{4}$$

$$= \mathbf{W}_e^\top\mathbf{Q}^\top\mathbf{Q}(\mathbf{w}_k - \mathbf{w}_j) \tag{5}$$

$$= \mathbf{W}_e^\top(\mathbf{w}_k - \mathbf{w}_j). \tag{6}$$

Thus the two models have the same input-space interference vector and therefore the same optimal perturbation direction. $\square$

Intuitively, rank or PCA determines the subspace in which attacks can act, but not the arrangement of meaningful feature directions inside that subspace. Superposition geometry is this internal arrangement, captured by the relevant Gram structure of class feature vectors. If two models learn the same geometry up to an orthogonal transformation, their inner products are preserved, so the perturbation that creates constructive interference in one model creates the same constructive interference in the other. This provides a mechanism for transferability.

## C. Synthetic argmax task

### C.1. Hypotheses testing framework

We explicitly state our hypotheses for the three research questions in Sec. 4.

*Research Q1*: Do adversarial perturbations exploit superposition geometry?

- $H_0$: Adversarial perturbations are random with respect to feature geometry.
- $H_1$: Adversarial perturbations systematically exploit geometric relationships between superposed representations.

*Research Q2*: Do data correlations determine superposition geometry?

- $H_0$: Input correlations have no systematic effect on learned geometries.
- $H_1$: Input correlations determine geometric arrangements across model initializations.

*Research Q3*: Does shared geometry explain attack transferability?

- $H_0$: Attack transferability is independent of geometric similarity.
- $H_1$: Transferability increases with shared latent structure.

We test these hypotheses through controlled experiments:

- $H_1(1)$: We measure the input perturbation profile alignment with a class's latent representation and the latent attack vector.
- $H_1(2)$: We systematically vary input correlations and measure resulting geometries.
- $H_1(3)$: We quantify transferability rates across models with varying geometric similarity.

### C.2. Non-linear toy model

To verify that the interference-driven attack mechanism generalises beyond the strictly linear bottleneck used in the main text, we test a non-linear variant of the toy classifier that places a ReLU activation and bias on the decoder, following the architectural convention of Elhage et al. (2022):

$$\mathbf{h} = \mathbf{W}_e \mathbf{x},$$
$$\mathbf{z} = \mathrm{ReLU}(\mathbf{W}_d \mathbf{h} + \mathbf{b}_{\mathrm{dec}}).$$

The bottleneck $\mathbf{h}$ remains linear, so the class feature vectors $\mathbf{v}_c$ retain the same geometric interpretation as in the main text, and Proposition 1 continues to specify the optimal perturbation.

**Setup**. We use the same task, training procedure, and attack methodology as the main text (untargeted $\ell_2$-PGD at $\epsilon = 0.2$). We do not report the low dimensional $k = 6$ result due to the ReLU exhibiting a *dead-class* initialisation pathology (nearly every sample has a single active input feature so each class's pre-logit sign is determined by initialisation, and leading to dead ReLUs that never recover).

*Table 5.* Alignment between PGD and optimal perturbations for the non-linear variant ($\varepsilon = 0.2$). We report cosine similarity (mean $\pm$ std across 5 seeds) of the per-seed mean. All are highly significant (paired $t$-test comparing PGD vs. random direction across seeds).

| $k$ | $m$ | Accuracy | Sim. (PGD vs. Theory) | Sim. (Random) |
|---|---|---|---|---|
| 30 | 10 | $0.94 \pm 0.04$ | $0.96 \pm 0.00$ | $0.00 \pm 0.00$ |
| 90 | 30 | $0.97 \pm 0.00$ | $0.92 \pm 0.00$ | $0.00 \pm 0.00$ |

**Results.** Tab. 5 reports per-seed cosine alignment between PGD-discovered perturbations and the Proposition 1 direction, using the same per-seed aggregation and paired $t$-test against a random-direction baseline as the main Table 1. Alignment with the linear theoretical prediction matches the main-text linear result.

## C.3. Correlations constrain arrangements

Tab. 6 extends our analysis of how input correlations shape feature geometry (Sec. 4.2) to a wider range of class counts ($k$) and hidden dimensions ($m$). For a model with $k$ class feature vectors, let $\mathbf{f}^{(i)} \in \mathbb{R}^{k(k-1)/2}$ be the upper-triangular off-diagonal entries of its $k \times k$ class-vector cosine similarity matrix. For each correlation regime we train $N = 10$ models and report the mean $\pm$ std dev of the Pearson correlation $\rho_{ij} = \text{Pearson}(\mathbf{f}^{(i)}, \mathbf{f}^{(j)})$ over all $\binom{N}{2} = 45$ unordered seed pairs. Cosines are rotation-invariant, so $\rho_{ij}$ measures whether the pairwise relationships between specific class *labels* agree across models. Pairwise comparisons across regimes use a two-sample $t$-test with Bonferroni correction.

*Table 6.* Geometric similarity results across correlation types and superposition configurations.

| Correlation Type | $k$ | $m$ | Geometric Similarity |
|---|---|---|---|
| Uncorrelated | 6 | 2 | $0.18 \pm 0.15$ |
| | 30 | 10 | $0.17 \pm 0.02$ |
| | 90 | 30 | $0.17 \pm 0.01$ |
| Paired | 6 | 2 | $0.47 \pm 0.07$ |
| | 30 | 10 | $0.26 \pm 0.07$ |
| | 90 | 30 | $0.25 \pm 0.01$ |
| Global | 6 | 2 | $0.92 \pm 0.04$ |
| | 30 | 10 | $0.88 \pm 0.01$ |
| | 90 | 30 | $0.80 \pm 0.01$ |

## C.4. Separating superposition effects from capacity reduction

To disentangle superposition pressure from capacity, we vary the number of classes $k$ across two bottleneck widths $m \in 2, 5$, comparing settings at matched superposition pressure ($k/m$) but different capacity. We report robust accuracy in Tab. 7. Reading down each column, increasing superposition pressure ($k/m$) at fixed capacity reduces robust accuracy; reading across each row, increasing capacity at fixed ($k/m$) improves it. These controls demonstrate that superposition pressure ($k/m$) is the driver of the adversarial vulnerabilities we observe, with greater capacity affording additional robustness at matched pressure.

*Table 7.* Robust accuracy (%) at $\varepsilon = 0.2$ as a function of superposition pressure ($k/m$) and bottleneck width ($m$). Reading *down* a column raises superposition pressure at fixed capacity; reading *across* a row raises capacity at fixed pressure.

| | **Robust Acc. (%)** | |
|---|---|---|
| $k/m$ | $m = 2$ | $m = 5$ |
| 1 | 98.7 | 92.0 |
| 2 | 75.8 | 87.2 |
| 3 | 31.0 | 59.0 |
| 4 | 11.1 | 33.5 |

## C.5. Rank alone does not determine attack structure

A low-rank account of adversarial vulnerability identifies the admissible feature subspace but not its internal geometry. Our superposition account predicts adversarial perturbations in our toy model as $\delta_i \propto \mathbf{v}_i^\top (\mathbf{v}_k - \mathbf{v}_j)$ (Prop. 1), which depends on the specific pairwise overlaps $\mathbf{v}_i^\top \mathbf{v}_j$ – the Gram structure inside the subspace – and is not invariant to rotations within it. As a result, two models sharing rank, latent dimension, and principal subspace can still exhibit different attack structures.

To make this concrete, consider two three-class models with $\mathbf{v}_i \in \mathbb{R}^2$.

**Model A**.
$$\mathbf{v}_1 = [1, 0]^\top, \quad \mathbf{v}_2 = [0, 1]^\top, \quad \mathbf{v}_3 = \tfrac{1}{\sqrt{2}}[1, 1]^\top.$$

For an attack from class 1 to class 2,
$$\delta_3 \propto \mathbf{v}_3^\top (\mathbf{v}_2 - \mathbf{v}_1) = \tfrac{1}{\sqrt{2}}(1, 1) \cdot (-1, 1) = 0,$$

so class 3 does not participate.

**Model B (same subspace, different arrangement)**.
$$\mathbf{v}_1 = [1, 0]^\top, \quad \mathbf{v}_2 = [0, 1]^\top, \quad \mathbf{v}_3 = \tfrac{1}{\sqrt{2}}[1, -1]^\top.$$

Now
$$\delta_3 \propto \mathbf{v}_3^\top (\mathbf{v}_2 - \mathbf{v}_1) = \tfrac{1}{\sqrt{2}}(1, -1) \cdot (-1, 1) = -\sqrt{2},$$

so class 3 participates strongly.

The two models share the same principal subspace and the same rank, but no orthogonal matrix $Q$ satisfies $\mathbf{v}'_i = Q\mathbf{v}_i$ for all $i$. The rank-constrained account therefore cannot distinguish them, yet they produce demonstrably different attacks. This complements Prop. 2, which establishes that orthogonal transformations *do* preserve optimal attacks. Our account can be viewed as a refinement of the low-rank account, with rank determining which subspaces are admissible, whilst the specific feature arrangement within that subspace determines which perturbations succeed.

### C.6. Additional examples of adversarial examples

Fig. 1 demonstrates how adversarial attacks exploit the interference between latent features in superposition. Here we provide further visual examples (Fig. 4) to reinforce intuition. Specifically, we supplement the main text by showcasing:

- Fig. 4a shows an additional instance of the setup in Sec. 4.1 ($m = 2$, $k = 7$) with an $\ell_2$-norm PGD attack, demonstrating the IPP and latent space manipulations that lead to misclassification.
- Fig. 4b shows a similar setup ($m = 2$, $k = 7$) using an $\ell_\infty$-norm PGD attack on a less sparse input.

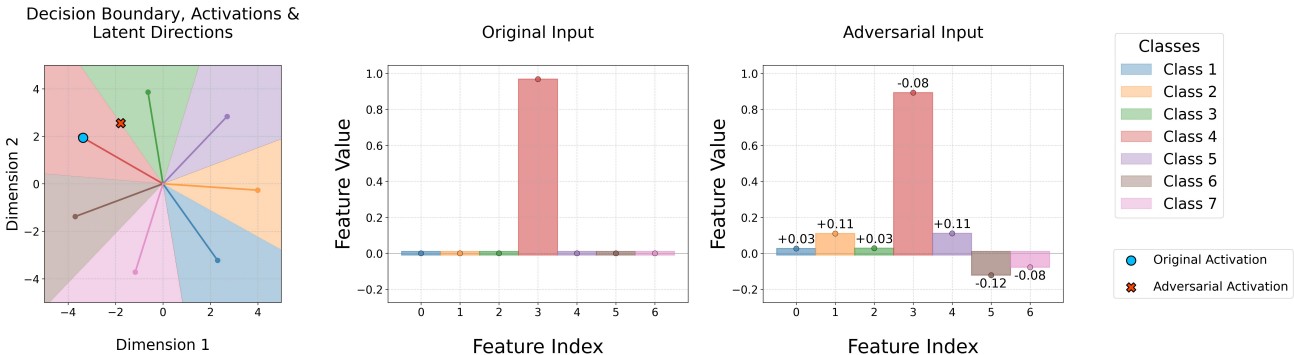

*(a)* An $\ell_2$-norm attack changing the classification of an input of class 4 to class 3. The left plot shows original and adversarial activations in latent space, along with representation directions. The right plots show original and perturbed input feature values.

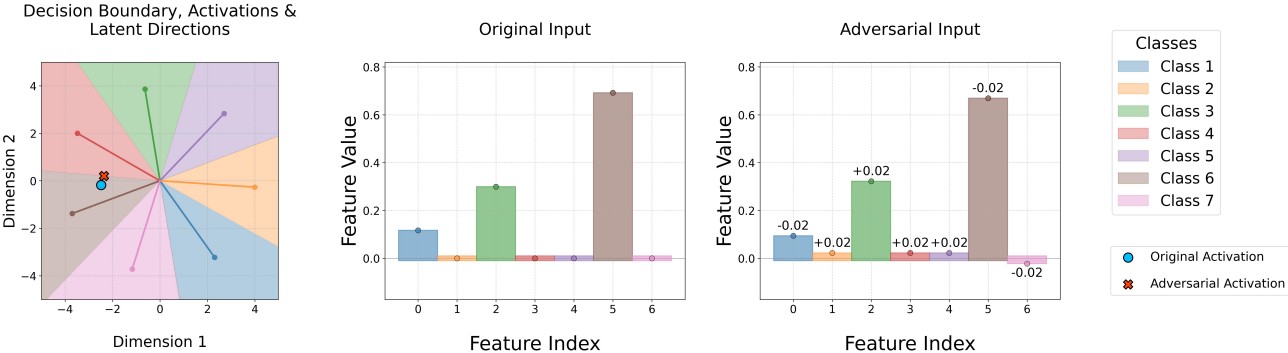

*(b)* An $\ell_\infty$-norm attack changing the classification of an input of class 6 to class 4. The left plot shows original and adversarial activations in latent space relative to class latent directions. The right plots show original and perturbed input feature values.

*Figure 4.* Visualisations of AEs in the toy model, supplementing Figure 1 from the main paper by illustrating attack mechanisms in activation space and input space under varied conditions.

### C.7. Generalisation across attack families

We focus on first-order gradient-based attacks throughout the main text, as they are the most widely used and practically relevant attacks in the literature. Prop. 1, however, derives the optimal perturbation from interference geometry independent of the attack

*Table 8.* **Alignment across attack families.** Cosine similarity between the attack perturbation and the theoretical optimum from Prop. 1 ($\epsilon = 0.1$). Mean $\pm$ std over $N$ attacks per configuration.

| Attack | Type | $(6, 2)$ | $(30, 10)$ | $(90, 30)$ |
|---|---|---|---|---|
| PGD-$\ell_2$ | 1st-order gradient | $0.93 \pm 0.01$ | $0.95 \pm 0.01$ | $0.97 \pm 0.00$ |
| SPSA | Zeroth-order | $0.92 \pm 0.01$ | $0.86 \pm 0.00$ | $0.79 \pm 0.00$ |
| Square | Score-based | $0.99 \pm 0.00$ | $0.84 \pm 0.00$ | $0.55 \pm 0.00$ |

algorithm, suggesting the mechanism may apply more broadly.

As an initial test, we replicate the attack-alignment experiment of Sec. 4.1 using two non-gradient-based families: SPSA (Uesato et al., 2018), a zeroth-order method that estimates gradients by finite differences, and Square Attack (Andriushchenko et al., 2020), a score-based gradient-free method. Tab. 8 reports cosine similarity to the theoretical optimum across all three attack families. We take this as preliminary evidence that the interference-based vulnerability structure we identify may not be specific to first-order attacks.

### C.8. Does our proposed mechanism persist at scale?

While empirical evidence suggests that larger models tend to be more robust to adversarial attacks, this effect is weak (Howe et al., 2025). When adversarial training is employed, clearer scaling trends emerge, but improvements remain largely specific to the attack type used during training rather than conferring general robustness.

Regarding superposition, we note a tension at play when scaling up models. Larger models have more capacity to represent concepts, but in general tasks such as next-token prediction over internet text there is a long tail of useful concepts to capture, so the number of features models must encode grows alongside their capacity. Despite this increased capacity, superposition appears to be prevalent even in frontier models (Lindsey et al., 2025). Supporting evidence comes from dictionary learning methods. SAEs require increasingly large dictionaries for larger models (Gao et al., 2025), suggesting that the number of features scales with model size. Hence, the fundamental tension driving superposition, that models must compress many features into limited dimensions, does not disappear with scale. Moreover, our own experiments (Sec. 5.3) show superposition emerging under standard weight-decay regularisation even without an explicit capacity bottleneck, and Prieto et al. (2026) argue that the resulting interference is a useful mechanism, not one that needs mitigation.

Since superposition persists in large-scale models, we conjecture that our insights remain relevant across model scales. Understanding how the superposition geometry changes with scale and with other realistic network components such as RMS norm, and whether the interference-based mechanism we identify governs adversarial perturbations in frontier networks, is interesting future work.

## D. Image classifier experiments

To assess whether the principles identified in the toy models extend to realistic settings, Sec. 5 reports experiments on four image datasets (CIFAR-10/100, STL-10, Tiny ImageNet) and three backbone architectures (ViT, ResNet-50, MLP-Mixer) with an engineered bottleneck head. This appendix gives the full architecture and training details (Sec. D.1) and presents extended results. We examine what shapes the learned superposition geometry in the vision setting, verify the trends hold across additional perturbation budgets, replicate on ResNet-50 and MLP-Mixer (Sec. D.4), and test reduced capacity as a confounder (Sec. D.3).

### D.1. Architecture & training information

We evaluate across four datasets and three backbone architectures. Unless stated otherwise, all configurations share the bottleneck head, training recipe, and augmentation pipeline described below.

**Datasets**. We use CIFAR-10 and CIFAR-100 (Krizhevsky, 2009) ($32 \times 32$, $k = 10$ and $k = 100$), STL-10 (Coates et al., 2011) ($96 \times 96$, $k = 10$), and Tiny ImageNet (Le & Yang, 2015) ($64 \times 64$, $k = 200$). For each dataset we hold out a stratified $10\%$ of the training set as a validation split.

**Backbones**. The base ViT (Dosovitskiy et al., 2020) comprised 12 transformer layers, embedding dimension $d = 384$, 6 attention heads per layer, and a per-block MLP hidden dimension of 1536, with learned positional embeddings and a CLS token. For CIFAR ($32 \times 32$) it used $4 \times 4$ patches; for Tiny ImageNet ($64 \times 64$) it used $8 \times 8$ patches, giving 64 patches in both cases. The ResNet-50 (He et al., 2016) backbone followed the standard torchvision architecture trained from scratch, with the CIFAR-style stem (the $7 \times 7$ stride-2 stem convolution replaced by a $3 \times 3$ stride-1 convolution and the initial max-pool removed) to preserve spatial resolution; global average pooling yields a 2048-dimensional feature. We use ResNet-50 for STL-10, whose $5{,}000$ labelled images are too few to train a $\sim$30M-parameter ViT from scratch. The MLP-Mixer (Tolstikhin et al., 2021) backbone (used on CIFAR) had hidden dimension 512, depth 8, $4 \times 4$ patches (64 tokens), token-mixing MLP dimension 256, and channel-mixing MLP dimension 2048, with mean pooling over tokens.

**Bottleneck**. The bottleneck head consisted of a linear encoder projecting the pooled backbone features into an $m$-dimensional latent space, followed by a linear decoder mapping to the $k$ class logits. We use $m \in \{2, 3, 5, 10\}$ for CIFAR-10 and STL-10, $m \in \{20, 30, 50, 100\}$ for CIFAR-100, and $m \in \{40, 60, 100, 200\}$ for Tiny ImageNet.

**Training**. The backbone and bottleneck were trained jointly end-to-end for up to 300 epochs with early stopping (patience 30–50). Training used the AdamW optimiser with learning rate 0.001 and weight decay 0.1, a batch size of 128, and a schedule of 10 epochs of linear warmup followed by cosine annealing to $10^{-6}$. The loss was cross-entropy with label smoothing 0.1. Dropout (0.1), Mixup ($\alpha = 0.8$), and CutMix ($\alpha = 1.0$) were used during training, and each configuration was repeated over five random seeds. For preprocessing, images were augmented with `RandomCrop` (padding 4 for CIFAR, 8 for Tiny ImageNet, 12 for STL-10), `RandomHorizontalFlip`, and RandAugment (2 operations, magnitude 9), then normalised to the per-dataset mean and standard deviation.

### D.2. Additional perturbation budgets

Tab. 9 extends the main paper analysis (Tab. 2) by evaluating the correlation between latent and input-space attack profiles across three perturbation budgets ($\epsilon \in \{2/255, 4/255, 8/255\}$), demonstrating that the relationship between bottleneck dimension and attack structure holds consistently across different attack strengths.

### D.3. Vulnerability is not reduced capacity

An alternative explanation for our results may be that compression simply reduces model capacity, and lower-capacity models are more fragile in general. We test against this by comparing adversarial robustness against robustness to common corruptions (Hendrycks & Dietterich, 2019) as the bottleneck varies (Tab. 10). We evaluate ViTs on CIFAR-10-C, which applies 15 common corruptions (e.g. blur, noise, weather effects) at varying severities.

Corruption robustness tracks clean accuracy: as the bottleneck tightens and clean accuracy falls, corruption accuracy falls with it (from 77.2% at $m/k = 1.0$ to 50.2% at $m/k = 0.2$). Adversarial robustness shows no such dependence – it remains uniformly low across all bottleneck ratios, including the widest. Were vulnerability a generic capacity effect, both measures would degrade together; instead only adversarial robustness is decoupled from capacity, consistent with a targeted geometric mechanism rather than a loss of representational headroom.

### D.4. Replication across architectures

Interference is not specific to the ViT architecture. Here we give the results for ResNet-50 (He et al., 2016) and MLP-Mixer (Tolstikhin et al., 2021), which differ substantially from ViTs in their inductive biases. For each we vary the bottleneck ratio $m/k$ and report clean accuracy, the mean latent–input profile correlation $\bar{\rho}$, and the interference magnitude $I$ (Sec. 5.1). In both architectures, decreasing $m/k$ increases $\bar{\rho}$ and $I$ monotonically, mirroring the ViT results in Tabs. 2 and 3.

### D.5. Correlations in class features

In Sec. 4 it was correlations between inputs that drove superposition arrangements. We note that in the vision classifiers it is not the correlations between input classes but rather the correlations in the representations at this point in the network that

*Table 9.* **Input class attack correlation across bottleneck dimensions and perturbation budgets**. $\bar{\rho}$ is the mean correlation between latent attack profile $\mathbf{r}$ and input-space attack profile $\mathbf{a}$ (excluding source/target).

| | CIFAR-10 | | | CIFAR-100 | |
|---|---|---|---|---|---|
| $\epsilon$ | $m$ | $\bar{\rho}$ | $\epsilon$ | $m$ | $\bar{\rho}$ |
| | 2 | 0.89 | | 20 | 0.61 |
| $2/255$ | 3 | 0.81 | $2/255$ | 30 | 0.48 |
| | 5 | 0.59 | | 50 | 0.42 |
| | 10 | 0.16 | | 100 | 0.34 |
| | 2 | 0.80 | | 20 | 0.53 |
| $4/255$ | 3 | 0.67 | $4/255$ | 30 | 0.44 |
| | 5 | 0.48 | | 50 | 0.43 |
| | 10 | 0.25 | | 100 | 0.37 |
| | 2 | 0.78 | | 20 | 0.49 |
| $8/255$ | 3 | 0.64 | $8/255$ | 30 | 0.39 |
| | 5 | 0.44 | | 50 | 0.34 |
| | 10 | 0.30 | | 100 | 0.29 |

*Table 10.* **Adversarial vs. corruption robustness** (ViT, CIFAR-10 / CIFAR-10-C, $\ell_\infty$ PGD at $\epsilon = 2/255$). Corruption accuracy tracks clean accuracy as the bottleneck varies, whereas adversarial accuracy stays uniformly low, indicating vulnerability is not a generic consequence of reduced capacity. Mean $\pm$ std over 5 seeds.

| $m/k$ | Clean Acc. | Adv. Acc. | Corrup. Acc. |
|---|---|---|---|
| 0.2 | $57.6 \pm 2.2$ | $11.5 \pm 0.7$ | $50.2 \pm 2.0$ |
| 0.3 | $78.2 \pm 1.4$ | $12.9 \pm 0.7$ | $66.5 \pm 1.2$ |
| 0.5 | $87.5 \pm 0.5$ | $13.4 \pm 0.7$ | $74.8 \pm 0.7$ |
| 1.0 | $89.1 \pm 0.9$ | $13.8 \pm 0.8$ | $77.2 \pm 0.6$ |

*Table 11.* **ResNet-50** results. $\bar{\rho}$: mean latent–input profile correlation; $I$: mean absolute interference. Mean $\pm$ std over 5 seeds.

| Dataset | $m/k$ | Acc. | $\bar{\rho}$ | $I$ |
|---|---|---|---|---|
| CIFAR-10 | 0.2 | $94.5 \pm 0.9$ | $0.80 \pm 0.02$ | $0.60 \pm 0.00$ |
| | 0.3 | $96.6 \pm 0.2$ | $0.69 \pm 0.01$ | $0.47 \pm 0.00$ |
| | 0.5 | $96.3 \pm 0.6$ | $0.46 \pm 0.02$ | $0.26 \pm 0.01$ |
| | 1.0 | $97.2 \pm 0.1$ | $0.33 \pm 0.02$ | $0.12 \pm 0.00$ |
| CIFAR-100 | 0.2 | $83.1 \pm 0.2$ | $0.38 \pm 0.01$ | $0.16 \pm 0.00$ |
| | 0.3 | $83.2 \pm 0.1$ | $0.31 \pm 0.01$ | $0.12 \pm 0.00$ |
| | 0.5 | $83.1 \pm 0.3$ | $0.22 \pm 0.01$ | $0.08 \pm 0.00$ |
| | 1.0 | $83.2 \pm 0.2$ | $0.15 \pm 0.01$ | $0.05 \pm 0.00$ |

*Table 12.* **MLP-Mixer** results. $\bar{\rho}$: mean latent–input profile correlation; $I$: mean absolute interference. Mean $\pm$ std over 5 seeds.

| Dataset | $m/k$ | Acc. | $\bar{\rho}$ | $I$ |
|---|---|---|---|---|
| CIFAR-10 | 0.2 | $82.7 \pm 0.9$ | $0.70 \pm 0.04$ | $0.61 \pm 0.00$ |
| | 0.3 | $89.8 \pm 0.2$ | $0.49 \pm 0.04$ | $0.46 \pm 0.00$ |
| | 0.5 | $90.9 \pm 0.4$ | $0.33 \pm 0.04$ | $0.22 \pm 0.01$ |
| | 1.0 | $91.5 \pm 0.3$ | $0.15 \pm 0.03$ | $0.12 \pm 0.00$ |
| CIFAR-100 | 0.2 | $66.7 \pm 1.1$ | $0.50 \pm 0.01$ | $0.16 \pm 0.00$ |
| | 0.3 | $62.5 \pm 2.5$ | $0.42 \pm 0.08$ | $0.12 \pm 0.00$ |
| | 0.5 | $67.7 \pm 0.4$ | $0.25 \pm 0.02$ | $0.08 \pm 0.00$ |
| | 1.0 | $68.1 \pm 0.8$ | $0.19 \pm 0.04$ | $0.04 \pm 0.00$ |

drive these arrangements. At the classification layer this is likely similar to the confusion matrix – *i.e.* how each class is misclassified in relation to the other classes. Classes that cluster together are those the network finds inherently similar and misclassifies together. To test this we repeat the experiment, finetuning the base ViT using timm/vit_base_patch16_384 which has been trained on ImageNet-21k (14 million images, 21,843 classes) and ImageNet (1 million images, 1,000 classes). After fine-tuning on CIFAR-10 and applying the same bottleneck training procedure as Sec. 5, the resulting geometry shows random ordering between initialisations with approximately equal spacing between features (*i.e.* neural collapse (Kothapalli, 2023)). In this case performance is near 100%, meaning the confusion matrix is the identity, and the superposition loses its structure. In contrast, models trained solely on CIFAR-10 converge to similar geometries because they share the same learned difficulty structure - the same pairs of classes prove challenging to distinguish, leading to consistent superposition patterns.

## E. Inspecting adversarial perturbations via SAE features

The analyses of the main paper focused on scenarios with either explicitly defined class features (toy model) or class-level representations in a bottleneck (ViT experiments). However, in large, practical networks, interference will occur between unlabelled latent features across all layers. As mentioned in the future work section, SAEs offer a promising unsupervised method to extract more linear features from NNs (Bricken et al., 2023). This appendix section presents a preliminary investigation into using SAE-extracted features to characterise how a large scale ViT responds internally when processing AEs versus clean inputs. These initial results are intended to explore the feasibility of this approach for future work aimed at understanding adversarial phenomena on a feature level within large models, rather than presenting a conclusive new set of findings. We first outline the experimental setup and validate the SAEs behaviour on adversarial inputs before presenting exploratory layer-wise feature difference analyses.

### E.1. Experiment setup

We analyse SAE-extracted features from the vision encoder of the LAION/CLIP-ViT-B-32-DataComp.XL-s13B-b90K model, a 12-layer ViT with $\approx 70\%$ zero-shot accuracy on ImageNet-1k (Deng et al., 2009). We use open-source Prisma SAEs mapping the 768-dimensional activation space to a dictionary of 49,152 features ($64\times$ expansion) (Joseph et al., 2025). We examine SAEs trained on either mean patch token representations or CLS token representations, across all layers. We analyse features activating above a high threshold (0.1) to capture changes to the highest activating features, and a low threshold (0.001). We generate 10,000 original/successfully-attacked image pairs ($\ell_\infty$-norm, $\epsilon = 0.02$) for our SAE feature comparisons. This setup allows for an initial exploration of how adversarial perturbations manifest at the level of SAE features.

### E.2. SAE validity for adversarial inputs

Given that AEs are out-of-distribution inputs, and our aim is to use standardly trained SAEs to understand them, we first conduct a basic validation. This subsection examines the impact on model accuracy and SAE reconstruction when SAEs are inserted at inference time for both clean and adversarial inputs. The validation has two goals: (i) confirm that SAE reconstruction is faithful enough to preserve the model's clean-input behaviour, and (ii) confirm that the SAE does not filter out the adversarial signal, *i.e.* the attack still succeeds when run through the SAEs. Together these ensure that subsequent feature-level differences reflect real attack-induced changes rather than artefacts of SAE reconstruction.

*Table 13.* Impact on ImageNet-1k classification accuracy when inserting a pre-trained SAE (one layer at a time) during inference for original, $\ell_\infty$ adversarial ($\epsilon = 0.02$), and random noisy inputs. 'Accuracy' refers to the ViT's top-1 accuracy. 'Original input + SAE' means the clean ViT activations at a given layer are passed through the SAE then its reconstruction is passed to the next layer. The slight 'de-attacking' effect (improved accuracy for 'Adv. input + SAE' vs 'Adv. input') suggests SAEs might filter some adversarial noise, but overall, the model still largely misclassifies, indicating salient adversarial changes for misclassification are processed.

| Metric | Layer 2 | Layer 5 | Layer 8 | Layer 11 |
|---|---|---|---|---|
| **Accuracy** | | | | |
| Original input | 1.00 | 1.00 | 1.00 | 1.00 |
| Original input + SAE | 0.99 | 0.99 | 0.99 | 0.86 |
| Adv. input | 0.00 | 0.00 | 0.00 | 0.00 |
| Adv. input + SAE | 0.02 | 0.01 | 0.01 | 0.11 |
| Noisy input | 0.98 | 0.98 | 0.98 | 0.98 |
| Noisy input + SAE | 0.98 | 0.98 | 0.98 | 0.85 |
| **SAE Improvement** | | | | |
| Original input | -0.01 | -0.01 | -0.01 | -0.14 |
| Adv. input | 0.02 | 0.01 | 0.01 | 0.11 |
| Noisy input | -0.00 | -0.00 | -0.00 | -0.13 |

### E.3. Layer-wise SAE feature differences

This subsection presents exploratory visualisations comparing SAE feature activations across different layers of the ViT for original, $\ell_\infty$ adversarial, and random noisy inputs. These visualisations are intended to highlight potential avenues for future, more in-depth investigation into how adversarial attacks distinctively alter sparse feature representations. We examine:

- Activation count: Histograms comparing the number of SAE features activated above a set threshold in a layer for original, adversarial, and noisy images. This explores whether attacks characteristically alter feature sparsity.
- Feature overlap: The degree of overlap (number of common activated features) between (original vs. adversarial) and (original vs. noisy) conditions. This explores if attacks activate a distinctly different set of features.
- Jaccard Similarity: The Jaccard similarity between sets of activated features, offering a normalised measure of overlap.
- Mean activation: Histograms of mean activation values for features active above a threshold. This looks for changes in the intensity of (key) features.
- Distinct Features: The count of unique features activated across an image set for each condition.

### E.4. CLS token SAE features with high activation threshold (0.1)

This subsection presents a preliminary analysis of SAE features derived from CLS tokens, only considering features activated above a threshold of 0.1. Figures 5 through 8 visualise and compare the histogram of activation counts, feature overlap, Jaccard similarity, and histogram of mean activations, respectively, for original, adversarial, and random noisy inputs across different ViT layers.

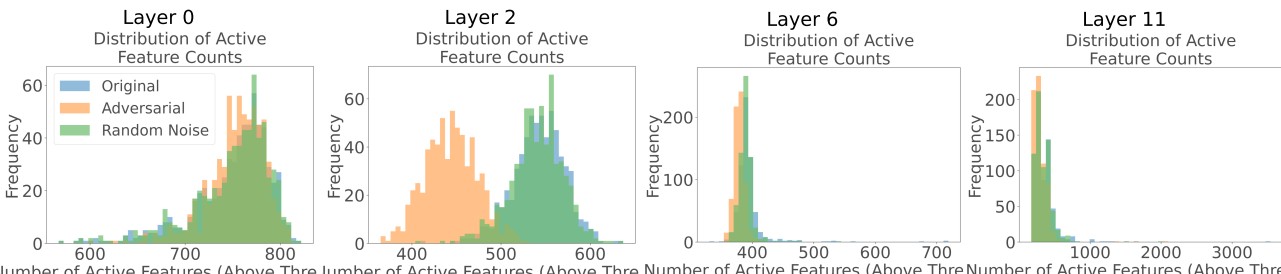

*Figure 5.* Histogram of activation counts for CLS token SAE features (threshold 0.1).

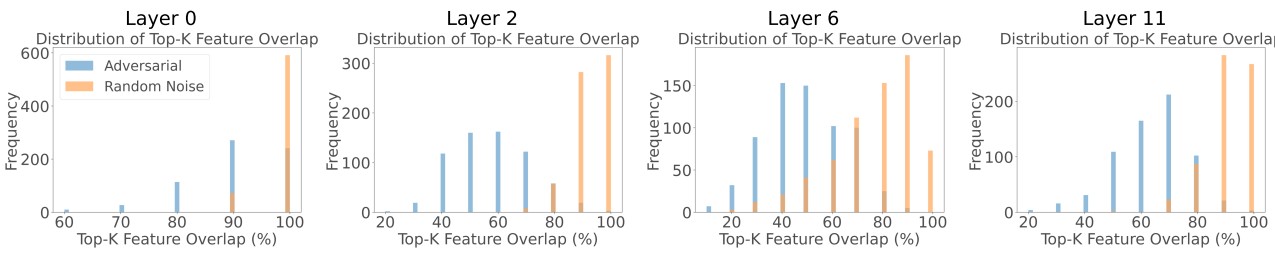

*Figure 6.* Feature overlap between original vs. attacked and original vs. noisy images for CLS token SAE features (threshold 0.1).

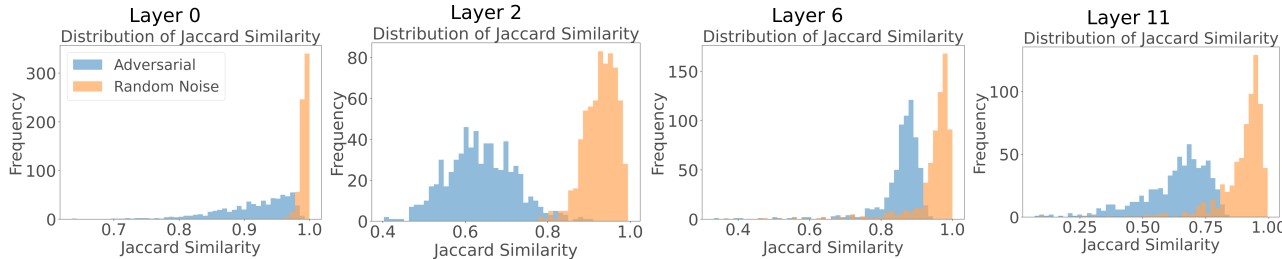

*Figure 7.* Comparison of Jaccard similarity for CLS token SAE features (threshold 0.1).

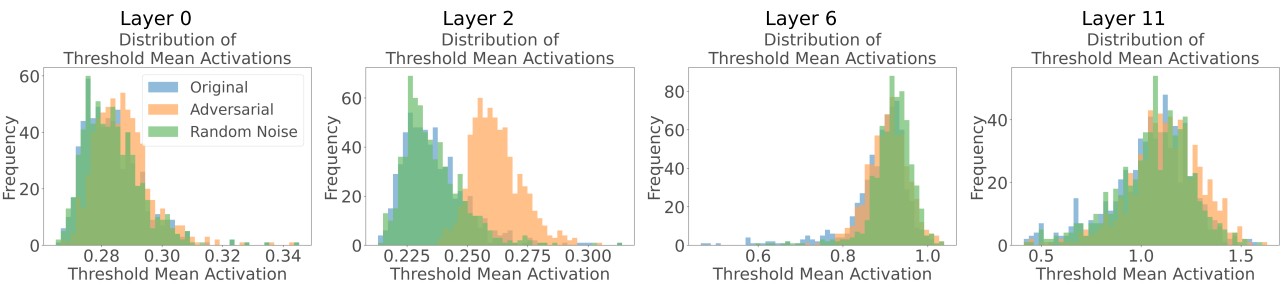

*Figure 8.* Histogram of mean thresholds for CLS token SAE features (threshold 0.1).

### E.5. CLS token SAE features with low activation threshold (0.001)

We examine SAE features from CLS tokens with a lower activation threshold of 0.001, to not just capture the most intensely activated features. The subsequent figures illustrate the histogram of activation counts (Fig. 9), feature overlap (Fig. 10), Jaccard similarity (Fig. 11), and histogram of mean activations (Fig. 12) across layers for original, adversarial, and noisy inputs.

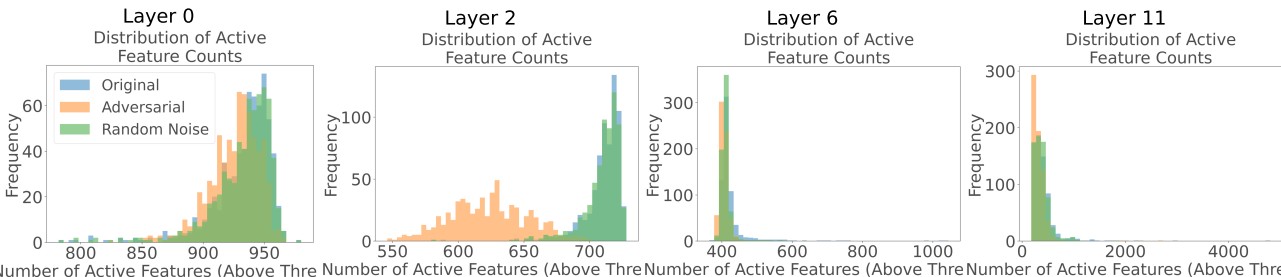

*Figure 9.* Histogram of activation counts for CLS token SAE features (threshold 0.001), with noise.

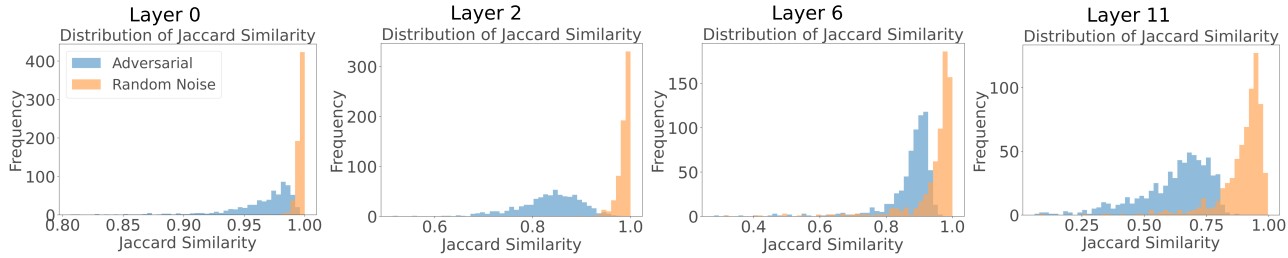

*Figure 10.* Feature overlap between original vs. attacked and original vs. noisy images for CLS token SAE features (threshold 0.001).

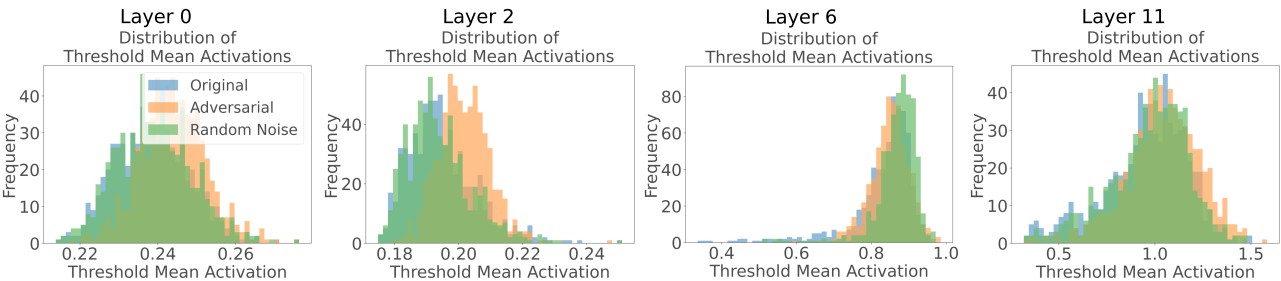

*Figure 11.* Comparison of Jaccard similarity for CLS token SAE features (threshold 0.001).

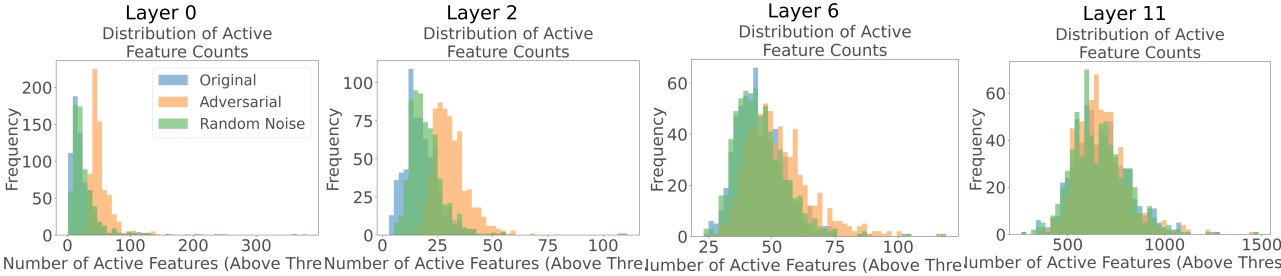

*Figure 12.* Histogram of mean activations for CLS token SAE features (threshold 0.001), with noise.

## E.6. Patch token SAE features with high activation threshold (0.1)

This subsection shifts the focus to SAE features derived from the mean of patch tokens, again using a high activation threshold of 0.1. We present comparisons of activation count histograms (Fig. 13), distinct feature count histograms (Fig. 14), feature overlap (Fig. 15), Jaccard similarity (Fig. 16), and mean activation histograms (Fig. 17) for original, adversarial, and noisy inputs.

*Figure 13.* Histogram of activation counts for patch token SAE features (threshold 0.1), with noise.

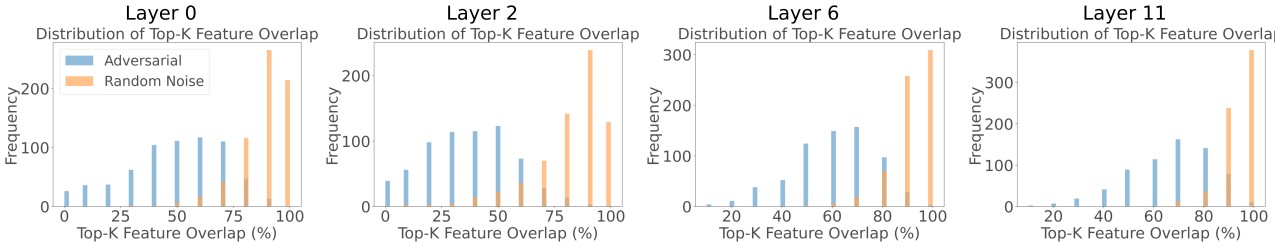

*Figure 14.* Histogram of distinct patch features for patch token SAE features (threshold 0.1), with noise.

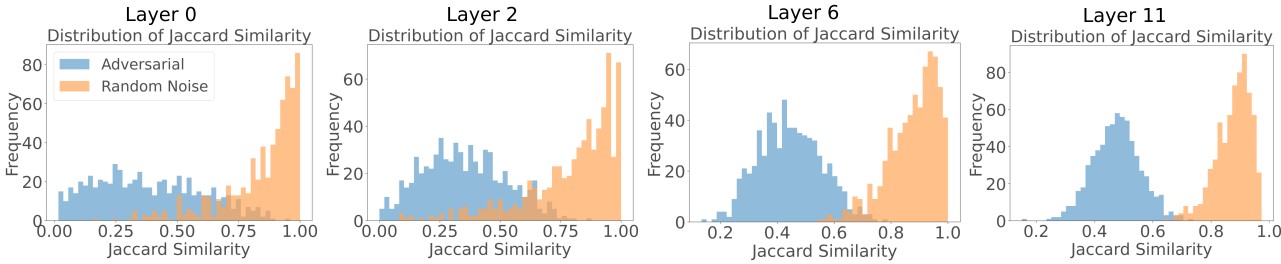

*Figure 15.* Feature overlap between original vs. attacked and original vs. noisy images for patch token SAE features (threshold 0.1).

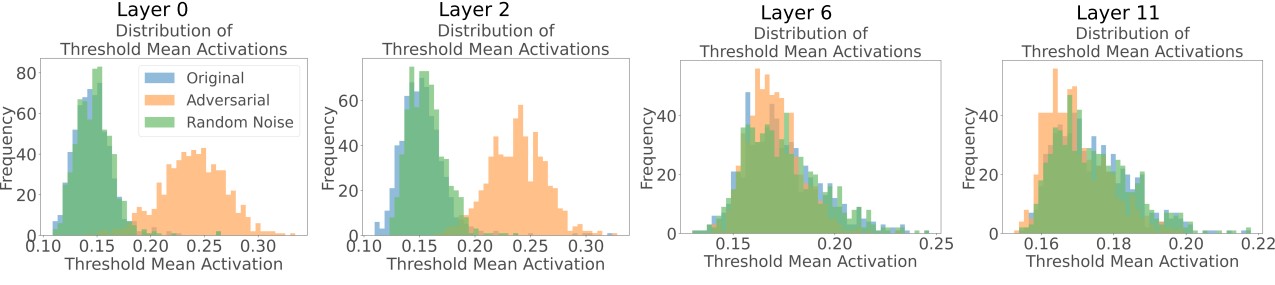

*Figure 16.* Comparison of Jaccard similarity for patch token SAE features (threshold 0.1).

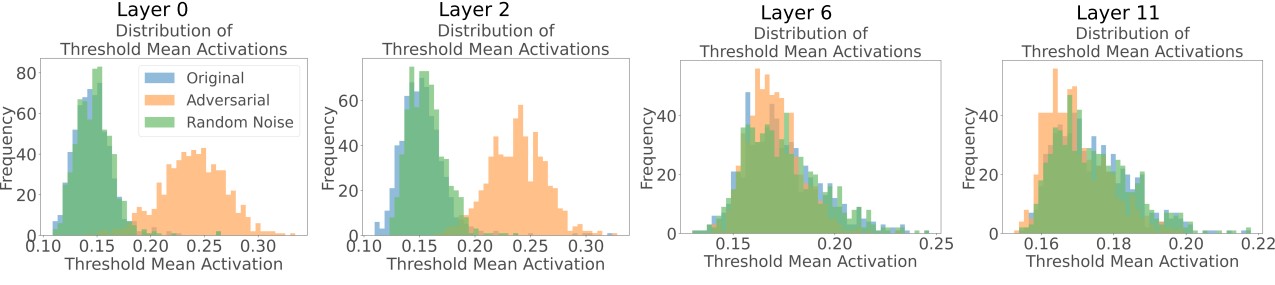

*Figure 17.* Histogram of mean thresholds for patch token SAE features (threshold 0.1), with noise.

### E.7. Patch token SAE features with low activation threshold (0.001)

Finally, this subsection explores SAE features from the patch tokens means using a low activation threshold of 0.001. The figures show histograms of activation counts (Fig. 18), distinct feature counts (Fig. 19), feature overlap (Fig. 20), Jaccard similarity (Fig. 21), and mean activation histograms (Fig. 22) for original, adversarial, and noisy inputs.

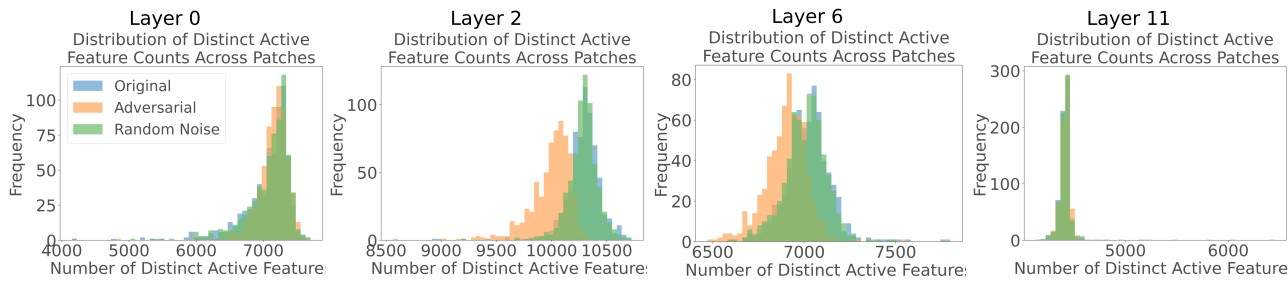

*Figure 18.* Histogram of activation counts for patch token SAE features (threshold 0.001), with noise.

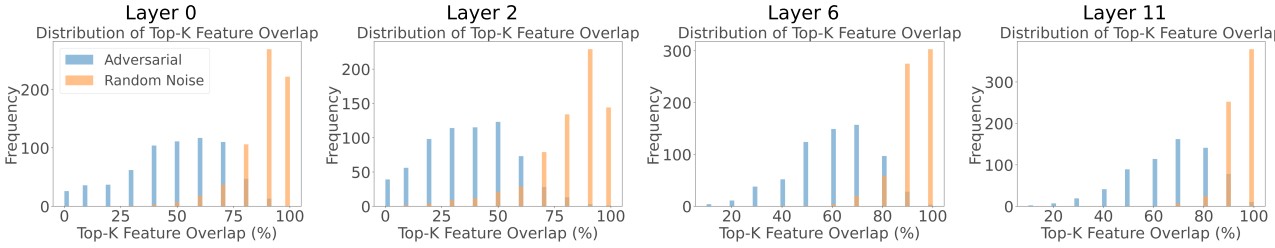

*Figure 19.* Histogram of distinct patch features for patch token SAE features (threshold 0.001), with noise.

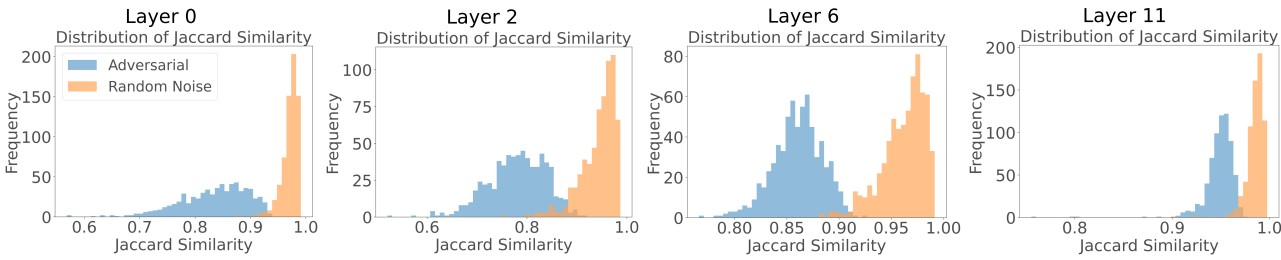

*Figure 20.* Feature overlap between original vs. attacked and original vs. noisy images for patch token SAE features (threshold 0.001).

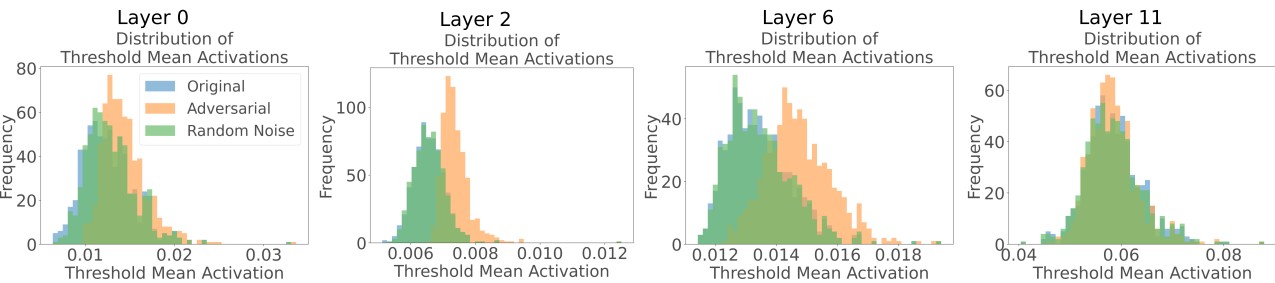

*Figure 21.* Comparison of Jaccard similarity for patch token SAE features (threshold 0.001).

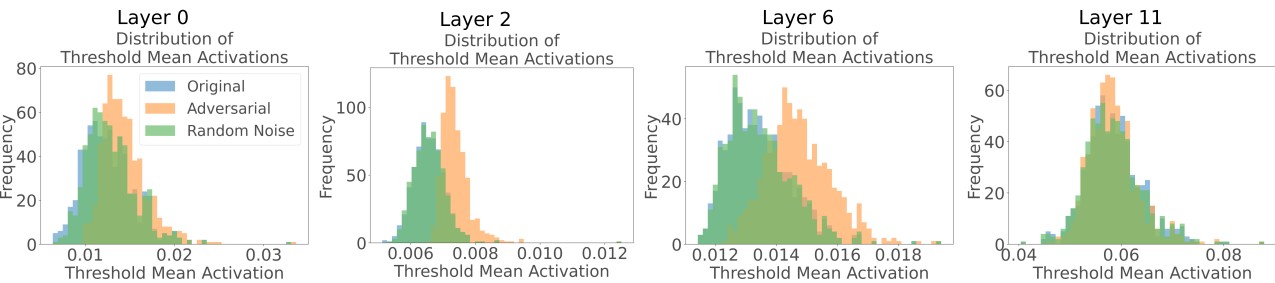

*Figure 22.* Histogram of mean activations for patch token SAE features (threshold 0.001), with noise.

