# OpenReview forum: "Adversarial Vulnerability from Interference Between Features in Superposition"
_ICML.cc/2026/Conference — ICML 2026 regular_

### Official Review · Reviewer_PQ28 · 2026-03-07

**Soundness:** 2
**Presentation:** 3
**Significance:** 2
**Originality:** 3
**Overall Recommendation:** 3
**Confidence:** 4

**Summary:**

This paper proposes a mechanistic account of adversarial vulnerability grounded in the phenomenon of superposition — the tendency of neural networks to represent more features than they have dimensions, resulting in non-orthogonal feature directions and inter-feature interference. The authors argue that gradient-based adversarial attacks (PGD) can exploit this interference geometry, that the structure of successful attacks is predictable from the geometry of the classification head, and that models trained on similar data develop similar interference patterns, explaining attack transferability. The theory is developed in a synthetic linear bottleneck setting where the optimal perturbation is derived in closed form, and validated empirically on ViT classifiers trained on CIFAR-10/100 with artificially engineered bottleneck layers. A supplementary experiment attempts to extend the findings to standard models via regularization-induced rank collapse.

**Compliance With Llm Reviewing Policy:**

Affirmed.

**Final Justification:**

The rebuttal addressed several of my original concerns. The new experiments across architectures (ResNet-50, MLP-Mixer) and datasets (TinyImageNet, STL-10, CIFAR-10-C) meaningfully strengthen the empirical scope by demonstrating that the observed trends hold beyond ViTs on CIFAR, though all experiments still operate within the explicit bottleneck regime rather than standard unconstrained models. The PAG citations and discussion resolve my related work concern, and I accept Gorton and Lewis (2025) as concurrent work. These updates led me to revise my scores upward.

However, my central concern (Q1) remains unresolved: the authors argued interpretively that superposition explains what fills the low-rank subspace, but did not provide a distinguishing experiment showing the superposition account produces qualitatively different predictions than rank-constrained backpropagation alone. I see merit in superposition as a complementary perspective and acknowledge the originality of this formalization, but remain uncertain whether the core mechanism is genuinely distinct from known gradient mechanics. I adjust my overall score from 1 to 3.

**Key Questions For Authors:**

**Q1.** How does the superposition-interference framework predict attack behavior differently from the simpler explanation that PGD perturbations follow gradients constrained by low-rank bottleneck structure (explicit or regularization-induced)? Can the authors provide additional theoretical analysis or experiments showing the superposition mechanism produces qualitatively different attack profiles than rank constraint alone?

**Q2.** Gorton & Lewis (2025) links superposition to vulnerability through geometric analysis and causal interventions across settings. What additional predictions or results does this paper's classification head formalization provide beyond their established superposition-vulnerability connection, particularly in realistic settings beyond bottlenecks?

**Q3.** The core empirical finding matches Perceptually Aligned Gradients (PAG) documented across multiple published works. What specific predictions does the superposition-interference framework make about attack structure that existing PAG/manifold geometry explanations do not cover? Can the authors provide concrete examples where their geometric account better explains or predicts PAG phenomena?

**Q4.** At realistic settings (m = k), input-latent attack profile correlations drop to ~0.16 (CIFAR-10, Table 2). Additionally, bottlenecked models exhibit low performance (58.7% accuracy at m/k=0.2, Table 3). What specific predictions does the framework make for standard non-bottlenecked models achieving competitive accuracy levels (>85%), and what experiments demonstrate applicability beyond artificial bottlenecks?

**Limitations:**

Partially addressed. The authors acknowledge reliance on simplified models and that superposition is sufficient but not necessary for vulnerability. However, the discussion omits the conflation of rank-constrained backpropagation with superposition effects, and lack of generalizability to standard models.

**Strengths And Weaknesses:**

### Strengths
The paper derives closed-form optimal perturbations from classification head geometry (Table 1), connecting interference to attack directions in bottleneck settings. Builds on PAG's gradient structure findings (Tsipras ICLR 2019; Ganz ICML 2023) and Gorton & Lewis's superposition-robustness link (NeurIPS 2025).

### Weaknesses
The central theoretical result is a trivial consequence of constrained backpropagation, not a discovery about superposition. The optimal perturbation — the transpose of the encoder matrix applied to the decision boundary normal — is simply the loss gradient backpropagated through a rank-constrained linear layer. When m = 2 and k = 10, the chain rule forces any gradient optimizer to concentrate perturbations in two dimensions. This holds regardless of whether the rank constraint arises from an explicit architectural bottleneck or from regularization-induced rank collapse (Section 5.3): in both cases, the gradient is geometrically constrained by the low-rank structure of the weight matrix, and PGD will inevitably concentrate perturbations along the dominant directions of that matrix. The superposition framing adds no explanatory power over this simpler account, and the authors do not propose any practical defense, attack strategy, or diagnostic tool that generalizes beyond this specific constructed setting — which would be the expected payoff of a genuinely mechanistic explanation.

The paper's central thesis—that superposition contributes to adversarial vulnerability—is not novel. Gorton & Lewis (2025, NeurIPS Mechanistic Interpretability Workshop) establishes this link through geometric superposition analysis and causal interventions across settings, including toy models and larger architectures. This paper's classification head formalization represents a specific case within this established superposition-vulnerability connection, yet cites Gorton & Lewis only peripherally in related work (Sec. 6: "investigate the link... interventions control vulnerability") rather than engaging their direct relevance to the core claim.

The paper's core empirical observation — adversarial perturbations exhibit predictable, semantically structured profiles across classes — precisely characterizes Perceptually Aligned Gradients (PAG), first identified by Tsipras et al. and Santurkar et al. (ICLR 2019), shown to be bidirectionally linked to robustness by Ganz et al. (ICML 2023), and explained via data manifold geometry by Srinivas et al. (NeurIPS 2023). The PAG literature anticipates mechanistic accounts of structured attack gradients like the paper's superposition-interference framework. Yet the reviewed paper cites none of this published work and presents these structured profiles — such as truck-to-car attacks predictably suppressing horse while amplifying ship logits — as novel consequences of classification head interference. Without engaging the PAG literature, the marginal contribution of the superposition-interference account over existing explanations remains unclear.

All other technical issues compound these fundamental flaws: strong empirical results are confined to artificially degraded models (58.7% CIFAR-10 accuracy at m = 2) with negligible effects at realistic settings (m = k); the no-bottleneck experiment reaches only m = 6 at heaviest regularization; Table 2 lacks confidence intervals across 5 seeds; and the "preliminary" SAE appendix (86% reconstruction fidelity) supports no main claim. Critically, no defense, detector, or practical application generalizes beyond the artificial bottleneck regime.

---

> ### Author Rebuttal · Authors · 2026-03-31
>
> We find the reviewer's scores disproportionately harsh relative to the contributions of our paper. The unusually low ratings appear to stem from a reductive interpretation of both our theoretical contribution and empirical scope, rather than a balanced assessment of our work. We respond across four fronts:
>
> 1. We demonstrate why characterising our results as a downstream implication of rank-constrained optimisation is incorrect.
> 2. We highlight that Gorton & Lewis (2025) is a concurrent work that does not diminish the novelty of our contributions and more importantly, that our paper has distinct contributions.
> 3. We show in detail how the reviewer's PAG literature comparison conflates distinct research questions.
> 4. We address remaining minor concerns.
>
> Overall, we believe this review applies an unduly dismissive standard and does not accurately reflect the substance of the work. We kindly ask the reviewer to reconsider the evaluation in light of these clarifications, which we elaborate on in detail below.
>
> **Q1: Superposition-interference vs rank-constraint**
>
> A rank-constraint account predicts perturbations are confined to an $m$-dimensional subspace governed by dominant singular directions of $W$. It operates at a different level of description than ours: it does not answer what these directions represent, which concepts interfere, or why independently trained models converge to the same vulnerable directions. Low-rank constraints in non-linear models induce superposition, emerging from data correlations and implicit training biases. The low-rank view describes the constraint; the superposition view explains what fills it and which feature concepts interact, connecting vulnerability to interpretable data structure rather than properties of W alone.
>
> This is tested in Figure 2. The rank-constraint account predicts the degree of gradient concentration under correlated data, but not which specific directions are shared. Two models with identical effective rank but different interference geometry would have very different transferability. The superposition account predicts both: correlated data forces similar interference geometry, driving transferability from 18% to 94%.
>
> A related point: adversarial vulnerability is overdetermined - many perturbations cause misclassification - yet the superposition framework identifies which perturbations PGD finds and why, via interference structure predictable from data statistics. This accounts for the strong alignment with theoretically predicted directions.
>
> **Q3: Connection to PAG literature**
>
> The connection to PAG is more subtle than the review suggests. We have cited these works and added a discussion to the revised manuscript.
>
> **Key distinctions.** PAG [2,3,4,5] is a property of adversarially trained models, characterising how input gradients are perceptually meaningful in pixel space. Our framework operates in activation space, showing that data correlations, low-rank structure, and weight decay can enforce semantically meaningful feature geometry whose interference patterns are exploited by attacks.
>
> **What our framework predicts.**
> 1. Which specific classes a given attack activates/suppresses, predicted from latent representations alone, with no input examples required (e.g. truck→car suppresses horse, amplifies ship).
> 2. How data correlations and compression jointly determine feature arrangements.
> 3.  An activation-space account of how feature interference determines attack structure, applying layer-agnostically to latent representations.
> 4. Transferability governed by similarity of learned interference geometries across models.
>
> **Q4: Scope and generalisability**
>
> New results across architectures and datasets (Tables 1-4, reviewer YKj8), including ResNet-50 achieving strong clean accuracies.
> * Not confined to degraded models. The m=2 setting aids visualisation; results span m/k=0.2 to 1.0 with significant effects throughout. For CIFAR-100, accuracy varies only 3% across ratios while profile correlations remain significant. ResNet-50 results confirm predictions hold at competitive accuracies.
> * Confidence intervals for Table 2 are now provided (see reviewer YKj8).
> * No-bottleneck setting. Sec. 5.3 shows weight decay reduces intrinsic dimension, producing the same effects. Our main claim is that bottlenecks arise naturally in many architectures (LLM unembedding, ImageNet heads, QK/OV projections).
> * SAE appendix is explicitly exploratory, supporting the call for future work, demonstrating feasibility of extending to intermediate layers via dictionary learning.
> * On defences. Our paper delivers mechanistic understanding and testable predictions, consistent with the scope of many PAG works the reviewer cites.
>
> **References**
>
> [1] Gorton+'25 [2] Tsipras+'19 [3] Santurkar+'19 [4] Ganz+'23 [5] Srinivas+'23 [6] Elhage+'22

---

> > ### Author Rebuttal · Reviewer_PQ28 · 2026-04-04
> >
> > I thank the authors for their detailed rebuttal. I acknowledge the substantial additional experiments across architectures (ResNet-50, MLP-Mixer) and datasets (TinyImageNet, STL-10, CIFAR-10-C), which strengthen the empirical scope of the work. I also acknowledge that the PAG literature has been cited and discussed, and accept that Gorton and Lewis (2025) can be considered concurrent work. These updates resolve my concerns regarding Q2, Q3, and Q4, and I adjust my score from 1 to 3 accordingly.
> >
> > However, my core concern Q1 remains unresolved. The authors argue that superposition explains what fills the low-rank subspace and why models share vulnerable directions, but this distinction remains interpretive rather than empirical. The key transferability result (Figure 2) can be equally explained by models trained on correlated data learning similar principal components, causing rank-constrained gradients to naturally align -- without requiring the superposition framing. I originally requested experiments or theoretical analysis showing the superposition mechanism produces qualitatively different predictions than rank-constrained backpropagation alone, which was not provided. I can see merit in superposition as a complementary perspective, but remain uncertain whether it constitutes a genuinely distinct mechanism or a redescription of known gradient mechanics. Resolving this would require a distinguishing experiment, which is not easily addressed in a short rebuttal.
> > I adjust my score to 3.

---

> > > ### Author Response · Authors · 2026-04-06
> > >
> > > We thank the reviewer for the thoughtful follow-up, engaging deeply with our work, and articulating this alternative explanation clearly. We agree with the reviewer on both fronts: (1) *low-rank structure constrains gradients, and any gradient-based attack is subject to this constraint*, and (2) *models trained on correlated data can learn similar principal components, which causes gradients constrained to a low-rank subspace to align*. The latter mechanism plausibly explains some degree of transferability, and we do not claim otherwise. However, our central claim is that this account is not complete, and that superposition makes additional predictions that are not implied by PCA / rank-constrained gradients alone.
> > >
> > > In particular, the PCA / rank-constrained account does not determine the internal geometry of that subspace, and therefore cannot predict:
> > >
> > > (i) **Class-specific attack structure**: our framework predicts which non-target classes are amplified or suppressed in an attack via $\delta_i \propto v_i^{\top}(v_t - v_s)$ (e.g., truck→car suppresses horse, amplifies ship). A PCA-based explanation predicts alignment of dominant directions, but does not specify these class-resolved interference patterns.
> > >
> > > (ii) **Bases vs subspaces**: the rank/PCA account determines only the admissible subspace, whereas attack structure depends on the pairwise interference geometry. The superposition account depends on the specific basis geometry (pairwise overlaps $v_i^{\top} v_j$: the **Gram structure inside that subspace**). Adversarial perturbations depend on $\delta_i \propto v_i^{\top}(v_k - v_j)$, which is not invariant to rotations within the subspace.
> > >
> > > To see this concretely, consider two models A and B sharing the same rank, latent dimension, and principal subspace:
> > >
> > > **Model A:** $v_1 = [1, 0]^\top$, $v_2 = [0, 1]^\top$, $v_3 = \frac{1}{\sqrt{2}}[1, 1]^\top$.
> > >
> > > Attack $1 \to 2$: $\delta_3 \propto v_3^\top(v_2 - v_1) = \frac{1}{\sqrt{2}}(1,1) \cdot (-1,1) = 0$. Class 3 does not participate.
> > >
> > > **Model B:** same subspace, different arrangement: $v_3 = \frac{1}{\sqrt{2}}[1, -1]^\top$.
> > >
> > > Now $\delta_3 \propto v_3^\top(v_2 - v_1) = \frac{1}{\sqrt{2}}(1,-1) \cdot (-1,1) = -\sqrt{2}$. Class 3 participates strongly.
> > >
> > > Note that there is no orthogonal matrix $Q$ such that $v_i' = Qv_i$ for all $i$. Therefore, two models with identical principal subspaces but different internal feature arrangements can exhibit different attack structure. This is not captured by the reviewer's explanation. Superposition can be seen as a refinement of low-rank structure, explaining how features are arranged within that subspace. We have now added this description to our paper.
> > >
> > > We now provide three experiments to demonstrate this distinction.
> > >
> > > **Experiment 1: Identical subspace yet different feature geometry leads to different attacks.**
> > > Five ReLU AEs ($h = \text{ReLU}(W_\text{enc} x)$, $\hat{x} = W_\text{dec} h$) are constrained to share the exact same principal subspace (verified at 1.0 similarity via $W_\text{enc} = A \cdot P$ for fixed projection $P$, learnable $A \in \mathbb{R}^{m \times m}$) but differ in learned feature geometry within that subspace. If the PCA account were sufficient, these models should produce identical attack structures, yet across settings ($d = 20$-$100$, $m = 5$-$20$), cross-seed attack similarity is only 0.50-0.68, far below 1.0.
> > >
> > > **Experiment 2: Rank alone underdetermines which features interfere.**
> > > Five ReLU AEs (with ReLU moved to reconstruction, following [1]), with $m = 5$, $d = 20$, trained on identical data with block-structured correlations (three groups of three features with within-block correlation 0.8 and between-block correlation 0.3, remaining features independent), varying only the random seed. Models share rank but not subspace or feature geometry. Cross-seed attack similarity is effectively zero ($\rho = 0.03$), while our interference predictor ($v_c^\top v_t$) predicted each model's own attack profiles with $\rho = 0.86$.
> > >
> > > **Experiment 3: Word-level predictions on internet-scale language data.**
> > > The same architecture as Exp. 2, trained on bag-of-words representations of WikiText-103 (vocab size $= 10,000$, $m = 1,000$), as per [2]. Given an input where target word $t$ is correctly reconstructed, we add $k$ words to suppress its reconstruction. The interference predictor ranks candidates by $(W^\top W)_{t,i}$; across 600 trials this ranking matches the brute-force oracle, achieving 90% suppression at $k = 20$. A baseline adding the $k$ highest-encoder-norm words (maximally perturbing the latent space without target-specific directional information) achieves 0% suppression. The attack words reflect learned anti-alignment in encoder geometry (e.g. "hypothesis", "larvae" suppress "September"). The rank-constraint view by itself cannot make these predictions.
> > >
> > > **References**
> > >
> > > [1] Elhage et al., Toy Models of Superposition, 2022.
> > >
> > > [2] Prieto et al., From Data Statistics to Feature Geometry, ICLR, 2026.

---

### Official Review · Reviewer_kLE5 · 2026-03-11

**Soundness:** 3
**Presentation:** 3
**Significance:** 3
**Originality:** 3
**Overall Recommendation:** 4
**Confidence:** 3

**Summary:**

This paper proposes a mechanistic account of adversarial vulnerability based on feature superposition and feature interference. The main idea is that when many features are represented in a limited-dimensional space, non-orthogonal feature directions create interference, and gradient-based adversarial perturbations exploit this structure. The paper investigates this hypothesis through a combination of analytically tractable synthetic tasks and experiments on bottlenecked vision transformers trained on CIFAR-10 and CIFAR-100, and further examines a setting without an explicit bottleneck by reducing effective dimensionality through regularization.

**Compliance With Llm Reviewing Policy:**

Affirmed.

**Final Justification:**

After considering the paper and the rebuttal together, I maintain my weak accept recommendation. What I find most compelling is not any single added experiment, but that the rebuttal sharpened the paper’s actual contribution: it presents a coherent mechanistic account linking superposition, interference, and adversarial vulnerability, and supports that account with a reasonably aligned chain of theory and evidence. The broader experiments and revised statistical treatment make the empirical case more credible, and the revised framing appropriately limits the strongest claims to settings the paper directly supports. I still view the no-explicit-bottleneck analysis as less clean than the explicit bottleneck results, but at this stage that is a limitation of scope rather than a flaw in the core argument. Overall, the rebuttal did not fundamentally change my view so much as increase my confidence that the paper’s central contribution is real, well-motivated, and likely to be useful to the robustness community.

**Key Questions For Authors:**

1.The synthetic results are convincing and the bottlenecked ViT results are promising. Do the authors have preliminary evidence that the same relationship between interference structure and adversarial vulnerability also appears in other common architectures, such as ResNets or ConvNets?
2.In the no-explicit-bottleneck setting, can the authors discuss more carefully how much of the observed effect can be attributed specifically to reduced effective dimensionality, as opposed to other consequences of weight decay?
3.Since the paper focuses primarily on PGD-style attacks, how broadly do the authors expect the proposed mechanism to generalize across different attack families?

**Limitations:**

The strongest evidence comes from controlled synthetic experiments, while the more realistic experiments are limited to a relatively narrow family of models and datasets. In addition, the no-explicit-bottleneck analysis is suggestive rather than fully causal. These limitations somewhat constrain the breadth of the conclusions, but they do not undermine the core mechanistic contribution of the work.

**Strengths And Weaknesses:**

Strengths:
1.The paper addresses an important and longstanding question in deep learning robustness: why adversarial examples arise and why they transfer across models. The proposed explanation is mechanistic and conceptually clear.
2.The synthetic setup is a major strength. It is analytically tractable, allows direct control over representation bottlenecks and data correlations, and is well aligned with the paper’s central research question.
3.The work successfully connects theory and experiment. The comparison between predicted attack directions and empirically found PGD directions is particularly compelling and provides concrete support for the proposed mechanism.
4.The paper offers a unifying perspective linking data correlations, representation geometry, feature interference, and transferability. I found this framing novel and intellectually valuable.
5.The paper does make a reasonable effort to move beyond toy models by evaluating bottlenecked ViTs and by exploring the no-explicit-bottleneck setting.


Weaknesses
1.The empirical validation beyond the synthetic setting is still somewhat narrow, mostly focusing on bottlenecked ViTs on CIFAR-10/100. Broader architectural and dataset coverage would strengthen the claims.
2.The “no explicit bottleneck” results are suggestive but not fully clean causally, since weight decay may influence several factors besides effective dimensionality.
3.The statistical treatment could be more rigorous in places, especially regarding dependence between attacks generated from the same trained model.
4.The attack evaluations are centered mainly on PGD-style first-order attacks, so the scope of the conclusions may be narrower than the broad framing sometimes suggests.

---

> ### Author Rebuttal · Authors · 2026-03-31
>
> We thank the reviewer for their thoughtful feedback and questions, and provide detailed answers below.
>
> **Q1 / W1: Broader architectural and dataset coverage**
>
> We agree that validation across architectures and datasets strengthens our claims. We have extended experiments to ResNet-50 [1] and MLP-Mixer [2] architectures, which differ substantially in their inductive biases, and to new datasets: TinyImageNet [7] (more classes), STL-10 [6] (higher resolution), and CIFAR-10-C [5] (15 common corruptions e.g. blur). Tables 1-3 in our response to reviewer YKj8 show that across all architectures and datasets, the core predictions hold: as superposition pressure increases, interference I increases monotonically and profile correlation $\bar{\rho}$ increases monotonically.
>
> Table 4 (reviewer YKj8) shows that corruption robustness scales with clean accuracy as the bottleneck ratio varies, while adversarial robustness remains uniformly poor. This indicates that adversarial vulnerability is not simply a byproduct of reduced model capacity under compression.
>
> **Q2 / W2: Explicit bottleneck vs regularised bottleneck**
>
> The explicit bottleneck setting is our clean intervention that isolates superposition pressure. The no-bottleneck results in Sec. 5.3 provide corroborating evidence that bottlenecks can occur even when ambient dimensionality is higher. We emphasise that explicit bottlenecks are our primary evidence, and that such compression arises naturally in **LLM unembedding matrices** (\~4k dimensions, >128k tokens), **ImageNet classification heads** (\~512 dimensions, 1,000 classes), and **QK/OV projections** in attention heads.
>
> We have revised Sec. 5.3 to discuss this more explicitly, noting that weight decay may affect factors beyond effective dimensionality.
>
> **W3: Rigorousness of statistical tests**
>
> We thank the reviewer for this point and provide a more conservative methodology to ensure our conclusions properly consider dependence between attacks generated from the same trained model (for similarity results (Tab. 1), transferability rates (Sec. 4.3, Fig. 2), and correlations (Tab. 2)).
>
> Taking Table 1 as a concrete example: we first compute one mean cosine similarity per seed, then run the t-test with $n=5$ per $(k, m)$ configuration. We have rerun all three sets of results under this methodology and find that all conclusions remain strongly statistically significant. We also direct the reviewer to Appendix C.1, which contains our complete hypothesis testing framework.
>
> **Q3 / W4: Scope of attack evaluations**
>
> We focused on first-order gradient-based attacks as they are the most widely used and practically relevant attacks in the literature. That said, we acknowledge the reviewer's point and have revised the abstract, Sec. 4, and Sec. 7 to scope claims explicitly to gradient-based attacks.
>
> In terms of generalisation to different attack types, our Proposition 1 derives optimal perturbations from interference geometry independent of the attack algorithm, suggesting the mechanism could apply more broadly. To test this, we replicated the Sec. 4.1 attack-alignment experiment using SPSA (zeroth-order) & Square Attack (score-based, gradient-free).
>
> As Table 5 shows, perturbations from all attack families align with the theoretical interference direction at levels comparable to PGD. We include this in the appendix as preliminary evidence that the vulnerability structure we identify may not be specific to first-order attacks.
>
> | Attack | Type | (6, 2) | (30, 10) | (90, 30) |
> | --- | --- | --- | --- | --- |
> | PGD-L2 | 1st-order gradient | 0.93 ± 0.01 | 0.95 ± 0.01 | 0.97 ± 0.00 |
> | SPSA | Zeroth-order | 0.92 ± 0.01 | 0.86 ± 0.00 | 0.79 ± 0.00 |
> | Square | Score-based | 0.99 ± 0.00 | 0.84 ± 0.00 | 0.55 ± 0.00 |
>
> Table 5: Alignment across attack families (ε = 0.1)
>
> We hope our answers and substantial additional results have addressed the reviewer's concerns, and would allow them to consider increasing their score.
>
> **References**
>
> [1] He+'16 [2] Tolstikhin+'21 [3] Dosovitskiy+'21 [4] Krizhevsky+'09 [5] Hendrycks+'19 [6] Coates+'11 [7] Le+'15

---

> > ### Author Rebuttal · Reviewer_kLE5 · 2026-04-01
> >
> > Thank you for the detailed rebuttal. My main concerns have been adequately addressed. The additional experiments substantially strengthen the empirical support, and the clarification around explicit versus no-explicit bottleneck settings is helpful. The revised statistical treatment and added non-PGD attack results also improve the paper. I still view the no-explicit-bottleneck setting as somewhat less clean causally, but this no longer materially affects my overall assessment. Overall, the rebuttal strengthens the paper and addresses the key concerns in my original review.

---

> > > ### Author Response · Authors · 2026-04-06
> > >
> > > We are pleased that Reviewer kLE5's concerns have been fully resolved. As such, we kindly ask the reviewer to consider increasing their score accordingly. We would be happy to address any further suggestions or questions during the remaining discussion period.

---

### Official Review · Reviewer_YKj8 · 2026-03-13

**Soundness:** 4
**Presentation:** 4
**Significance:** 3
**Originality:** 4
**Overall Recommendation:** 5
**Confidence:** 4

**Summary:**

This paper proposes a mechanistic explanation for adversarial vulnerability based on feature superposition in neural network representations. The authors argue that when models encode more conceptual features than the dimensionality of their activation space allows, those features must be represented as non-orthogonal directions, causing interference between them. They show theoretically in a controlled synthetic setting that adversarial perturbations exploit this interference and that the optimal perturbation direction can be predicted from the geometry of feature vectors. Empirically, the authors demonstrate that PGD attacks align with these predicted directions and that models trained on correlated data develop similar feature geometries, explaining adversarial transferability. They also provide experiments on ViT trained on CIFAR-10/100 further show that the relationship between feature interference and attack structure in practical models.

**Compliance With Llm Reviewing Policy:**

Affirmed.

**Final Justification:**

This paper provides an in-depth analysis of how superposition contributes to adversarial vulnerability.  My initial concerns were mainly on motivating the degree of supervision and on limited scope of experiments in terms of architecture, which have all been addressed in the author's rebuttal.

**Key Questions For Authors:**

See weaknesses

**Limitations:**

Yes

**Strengths And Weaknesses:**

Strengths:
- Brings an interesting insight on the connection between superposition and vulnerability to attacks, which furthers our understanding of what causes the adversarial vulnerability.  This is a novel perspective that I have not seen examined in other works
- Rigorous experimentation: I thought the scope of experiments in this paper was quite good.  I appreciated how the authors started with a synthetic task for which they were able to theoretically derive the optimal attack perturbation that has a form related to superposition and provided experiments with the synthetic data to show alignment with the attacks generated via PGD and computed via the derived formulation.  The synthetic task also helped with establishing connections to transferability as well which was interesting.  The authors also provided experiments for image classification with a bottleneck and without a bottleneck to demonstrate that these observed trends can hold with more practical architectures (ViT) and datasets.
- Clear organization and figures with main points highlighted in takeaway boxes

Weaknesses:
- While the controlled bottleneck experiments provide an interesting demonstration of how superposition can induce adversarial vulnerability, it remains somewhat unclear to what extent this mechanism contributes to robustness in practical models, where the dimensionality of the learned feature representations typically exceeds the number of classes and weight decay is quite low and therefore does not naturally impose the same degree of superposition.
- It would be nice to see results for other model architectures like ResNet

---

> ### Author Rebuttal · Authors · 2026-03-31
>
> We thank the reviewer for their constructive feedback and provide detailed answers below.
>
> **Degree of superposition in practical models**
>
> We appreciate the question on practical relevance and the opportunity to clarify where the bottleneck conditions studied in our paper arise in real models. We point to several important network components where strong compression occurs, meaning the dimension of the representation is significantly smaller than the number of class feature vectors (or more generally, feature vectors).
> 1. The **unembedding matrix of an LLM** (e.g. Llama 3 8B: ~4k dimensions, >128k tokens, compression ratio >30:1)
> 2. In vision classification, **ImageNet** classification heads (strong models ~512 dimensions but 1,000 classes)
> 3. The **QK/OV projections in attention heads** constitute bottlenecks that force non-orthogonal feature encoding.
>
> These examples demonstrate that the bottleneck conditions we study arise naturally in widely-used architectures, making our controlled setup representative of practical models. We agree that extending to intermediate layers is important future work, which will clarify when this mechanism dominates and when it is secondary (l.434-438).
>
> We limit our analysis to _class_ feature vectors in our work as it enables clean analysis. They can be read directly from **W** without approximation (i.e. no SAEs or probes are required), have known ground-truth labels, and enable direct measurement of interference. The broader use of feature vectors in intermediate layers would require confounds that we did not want to introduce.
>
> **Results on other architectures and datasets**
>
> We thank the reviewer for suggesting validation across architectures. We have extended our experiments to ResNet-50 [1] and MLP-Mixer [2], which differ substantially from ViTs [3] in their inductive biases, and to Tiny ImageNet [7] (more classes), STL-10 [6] (higher resolution), and CIFAR-10-C [5] (15 common corruptions e.g. blur).
>
> Tables 1-3 show that across all architectures and datasets, the core predictions hold: as superposition pressure increases, interference $I$ increases monotonically, and profile correlation $\bar{\rho}$​ increases monotonically.
>
> Table 4 shows that corruption robustness scales with clean accuracy as the bottleneck ratio varies, while adversarial robustness remains uniformly poor regardless of superposition pressure. This indicates that adversarial vulnerability is not simply a byproduct of reduced model capacity under compression, since corruption robustness scales with clean accuracy while adversarial robustness does not.
>
> We hope our answers and substantial additional results have addressed the reviewer's concerns, and would allow them to consider increasing their score.
>
> |Dataset|m/k|Acc.|ρ̄|I|
> |---|---|---|---|---|
> |CIFAR-10|0.2|94.5 ± 0.9|0.80 ± 0.02|0.60 ± 0.00|
> ||0.3|96.6 ± 0.2|0.69 ± 0.01|0.47 ± 0.00|
> ||0.5|96.3 ± 0.6|0.46 ± 0.02|0.26 ± 0.01|
> ||1.0|97.2 ± 0.1|0.33 ± 0.02|0.12 ± 0.00|
> |CIFAR-100|0.2|83.1 ± 0.2|0.38 ± 0.01|0.16 ± 0.00|
> ||0.3|83.2 ± 0.1|0.31 ± 0.01|0.12 ± 0.00|
> ||0.5|83.1 ± 0.3|0.22 ± 0.01|0.08 ± 0.00|
> ||1.0|83.2 ± 0.2|0.15 ± 0.01|0.05 ± 0.00|
>
> Table 1: ResNet-50 results.
>
> |Dataset|m/k|Acc.|ρ̄|I|
> |---|---|---|---|---|
> |CIFAR-10|0.2|82.7 ± 0.9|0.70 ± 0.04|0.61 ± 0.00|
> ||0.3|89.8 ± 0.2|0.49 ± 0.04|0.46 ± 0.00|
> ||0.5|90.9 ± 0.4|0.33 ± 0.04|0.22 ± 0.01|
> ||1.0|91.5 ± 0.3|0.15 ± 0.03|0.12 ± 0.00|
> |CIFAR-100|0.2|66.7 ± 1.1|0.50 ± 0.01|0.16 ± 0.00|
> ||0.3|62.5 ± 2.5|0.42 ± 0.08|0.12 ± 0.00|
> ||0.5|67.7 ± 0.4|0.25 ± 0.02|0.08 ± 0.00|
> ||1.0|68.1 ± 0.8|0.19 ± 0.04|0.04 ± 0.00|
>
> Table 2: MLP-Mixer results.
>
> | Dataset       | m/k  | Acc.       | ρ̄          | I           |
> | ------------- | ---- | ---------- | ----------- | ----------- |
> | CIFAR-100     | 0.2  | 65.7 ± 0.8 | 0.53 ± 0.01 | 0.16 ± 0.00 |
> |               | 0.3  | 66.6 ± 0.4 | 0.46 ± 0.01 | 0.12 ± 0.00 |
> |               | 0.5  | 68.1 ± 1.3 | 0.39 ± 0.00 | 0.08 ± 0.00 |
> |               | 1.0  | 68.8 ± 0.6 | 0.34 ± 0.01 | 0.06 ± 0.00 |
> | Tiny ImageNet | 0.25 | 54.6 ± 0.4 | 0.52 ± 0.01 | 0.11 ± 0.00 |
> |               | 0.5  | 55.1 ± 0.9 | 0.40 ± 0.01 | 0.07 ± 0.00 |
> |               | 1.0  | 56.4 ± 0.4 | 0.33 ± 0.01 | 0.06 ± 0.00 |
> | STL-10        | 0.2  | 67.1 ± 0.4 | 0.83 ± 0.00 | 0.60 ± 0.00 |
> |               | 0.3  | 82.8 ± 0.5 | 0.74 ± 0.02 | 0.46 ± 0.01 |
> |               | 0.5  | 88.0 ± 0.3 | 0.54 ± 0.01 | 0.18 ± 0.01 |
> |               | 1.0  | 89.0 ± 0.2 | 0.18 ± 0.02 | 0.12 ± 0.00 |
>
> Table 3: ViT results across additional datasets.
>
> |m/k|Clean Acc.|Adv. Acc.|Corrup. Acc.|
> |---|---|---|---|
> |0.2|57.6 ± 2.2|11.5 ± 0.7|50.2 ± 2.0|
> |0.3|78.2 ± 1.4|12.9 ± 0.7|66.5 ± 1.2|
> |0.5|87.5 ± 0.5|13.4 ± 0.7|74.8 ± 0.7|
> |1.0|89.1 ± 0.9|13.8 ± 0.8|77.2 ± 0.6|
>
> Table 4: Adversarial vs corruption robustness (CIFAR-10-C, ViT).
>
> **References**
>
> [1] He+'16 [2] Tolstikhin+'21 [3] Dosovitskiy+'21 [4] Krizhevsky+'09 [5] Hendrycks+'19 [6] Coates+'11 [7] Le+'15

---

> > ### Author Rebuttal · Reviewer_YKj8 · 2026-04-03
> >
> > Thank you for the additional experiments, my concerns have been addressed and will raise my score to a 5.

---

### Decision · Program_Chairs · 2026-04-30

**Decision:**

Accept (regular)

**Comment:**

Having read the paper and reviews, I do think there are points in which the paper can be improved. In particular, the hypothesis could be more convincingly conveyed by evaluating on larger datasets with more diversity, like the fully ImageNet corpus. However, there is a limit to the utility of high dimensional datasets in adversarial literature as the attack space grows.

On the positive side, the paper offers a refreshingly different and new perspective on how one can view adversarial attacks, with considerable expositionary work to evidence their point of view. This is indeed a strong point raised by the accept voting reviews in favor of the article. As such, I agree: it is a highly novel perspective with sufficient evidence and clarity to be accepted, and an excellent opportunity to broadcast a new view at ICML.